# A Contrast Minimization Approach to Remove Sun Glint in Landsat 8 Imagery

Frank Fell

Informus GmbH, 13187 Berlin, Germany; fell@informus.de

**Abstract:** Sun glint, i.e., direct solar radiation reflected from a water surface, negatively affects the accuracy of ocean color retrieval schemes if entering the field-of-view of the observing instrument. Herein, a simple and robust method to quantify the sun glint contribution to top-of-atmosphere reflectances in the visible and near-infrared is proposed, exploiting concomitant observations of the sun glint's morphology in the shortwave infrared. The method, termed Glint Removal through Contrast Minimization (GRCM), requires high spatial resolution (ca. 10–50 m) imagery to resolve the sun glint's characteristic morphology, meeting additional criteria on radiometric resolution, signal-to-noise ratio, and temporal delay between the individual band's acquisitions. It has been applied with good success to a selection of cloud-free Landsat 8 Operational Land Imager (OLI) scenes, otherwise encompassing a wide range of environmental conditions in terms of observation geometry, glint intensity, water types, as well as aerosol and Rayleigh optical depths. GRCM is entirely image based and does not require ancillary information on the sea surface roughness or related parameters (e.g., surface wind), nor the presence of homogeneous clear water areas in the image under consideration. GRCM's limitations are discussed, and its potential for sensors other than OLI as well as applications beyond glint removal are sketched.

**Keywords:** ocean color; sun glint; atmospheric correction; Landsat 8

## 1. Introduction

In Earth observation, the term "glint" refers to specular reflection of direct (sun glint) or diffuse (sky glint) solar radiation. Sun glint frequently poses a problem in remote observations of aquatic ecosystems as it may outshine the water leaving radiance carrying the signal of interest over large areas, thereby "confusing" water constituent retrieval schemes. Illustrating the importance of the problem, several ocean-observing instruments on polar orbiting platforms have been equipped with mechanisms to reduce the exposure to sun glint, for example the currently operational Ocean, Land and Cloud Imager (OLCI) onboard the Sentinel 3 series of satellites, tilted 12.6° westward [1].

The intensity of sun glint is controlled by the presence of sea surface facets allowing reflection of direct solar radiation into the field-of-view of the observing instrument: the more likely the occurrence of facets with the required orientation, the more intense is the sun glint signal. The instantaneous distribution of the sea surface facets' orientation depends on multiple processes occurring at different spatial and temporal scales. The formation and orientation of small capillary waves is primarily driven by surface winds [2,3], but also depends on atmospheric stability [4] as well as water temperature and density [5]. These small capillary waves form upon the underlying swell modifying the orientation of surface facets. The sea surface roughness is further influenced by the presence of currents [6], internal waves [7], as well as upwelling or mixing of water masses [8]. Biological activity or oil slicks may lead to the creation of surface films which will damp surface roughness [9]. Other processes of potential relevance include the presence of slush, or sub-surface topography [10].

Sea surface facets constitute an interface between two dielectric media, i.e., (sea) water and air. Reflection and transmission of light at respectively through these facets depend on the ratio of the medias' refractive indices as well as the angle of light incidence and are quantified by the Fresnel equations [11]. As the refractive index of water depends on wavelength, salinity, and temperature (for example n $\cong$ 1.337 for sea water at 15.0 °C, 35.0 PSU, and 700 nm) [12], so do reflectance and transmittance of light at/through the air/sea interface. As shown by [13], the wavelength dependence of the refractive index of water leads to a significant increase of the glint reflectance from longer towards shorter wavelengths (see Section 2.5). Sea surface facets are most likely oriented at near-zero slopes [14]. Nadir-looking imagers are therefore predominantly affected by sun glint at rather small solar zenith angles, while wide-swath instruments may also be markedly affected by sun glint at larger solar zenith angles.

Significant efforts have been made over the last decades to establish procedures to remove sun glint from ocean color observations. A thorough review of the approaches available by 2009 is provided in [15]. Information on sun glint correction methods published thereafter can be found for example in [13,16].

Earlier attempts to estimate and remove the sun glint contribution from medium resolution (ca. 300–1000 m) ocean color imagery have combined statistical models of the sea surface facet orientation with radiative transfer calculations [17]. Such methods depend critically on the availability of concomitant external information, for example, on the wind field controlling the orientation of the sea surface facets [2]. This information is not always available at the required accuracy, geographic detail and temporal resolution which is why correction methods aiming at assessing the sun glint intensity from external information often have not provided the desired results. This is especially true for high resolution imagery where small-scale or short-term processes do not average out over the area represented by an individual pixel.

A different approach has been adopted by [18] and subsequently optimized by [19] for high resolution ($\leq$10 m) multispectral imagery by establishing image and channel specific linear relationships to estimate the sun glint contribution in the visible (VIS, ca. 0.4–0.7 µm) from concomitant observations in the near infrared (NIR, ca. 0.7–1.5 µm), assuming negligible sub-surface contribution at the latter wavelengths in clear waters. The linear coefficients are then obtained by statistical regression applied to water areas encompassing both glint and glint-free pixels. A similar regression-based approach has been applied by [20] to develop a simple empirical glint correction method from MODIS (250–1000 m) observations over the Gulf of Mexico, exploiting information in the NIR at 0.859 µm to estimate the sun glint contribution in the VIS at 0.469 µm, 0.555 µm, and 0.645 µm. A spectral matching technique (POLYMER, POLYnomial based algorithm applied to MERIS) to disentangle the sun glint contribution from the atmospheric and the in-water contributions has been implemented by [21]. POLYMER is based on an iterative optimization scheme relying on relatively simple models of the atmospheric and oceanic optical properties, and fundamentally differs from the previously presented methods in that it exploits the full spectral information at individual pixel level.

The advent of a new generation of space-borne imagers providing concomitant observations in the VIS, NIR, and shortwave infrared (SWIR, ca. 1.5–2.5 µm) at high spatial (ca. 30 m or higher) and radiometric (12 bit or better) resolution such as the Operational Land Imager (OLI) onboard Landsat 8 has opened the path for the development of optimized sun glint correction schemes. Due to the strong absorption of pure water, the water leaving radiance at SWIR wavelengths can be reasonably assumed negligible [22], such that the top-of-atmosphere (TOA) reflectance above water surfaces at such wavelengths can be approximated as consisting of contributions from surface and atmosphere only. [23] have applied the regression approach of [19] to a four-year time series of OLI imagery offshore French Guiana, making use of the SWIR channel B7 (2.2 µm) to estimate the actual glint distribution and applying an automated scheme to identify the homogeneous deep clear water areas encompassing both glint and glint-free pixels as required to calculate the

regression parameters. [13] have developed a sun glint correction method for the Multi-Spectral Imager (MSI) onboard the Sentinel-2 series of satellites, using atmospherically corrected observations in the SWIR to determine the bidirectional reflectance distribution function (BRDF) of the water surface, and a theoretically derived model of the BRDF spectral dependency to subsequently assess the glint contribution at shorter wavelengths.

In the present article, a new method to estimate the sun glint contribution to the TOA reflectance in OLI images is introduced, combining elements of previously published approaches with a novel way of identifying and quantifying sun glint. While a linear model as suggested in [18,19] is applied to estimate the sun glint in the VIS and NIR (further on referred to as VNIR) from concomitant observations in the SWIR, the model coefficients are estimated herein by exploiting the sun glint's characteristic morphology through a contrast minimization approach, hence the designation GRCM (Glint Removal through Contrast Minimization).

A number of aspects distinguish GRCM from the previously mentioned regression-based glint assessment schemes [18–20,23]. No selection of suitable subareas containing both glint- and non-glint-affected pixels is required, GRCM identifies the entire glint affected area (GAA) by applying an automated analysis of the local reflectance contrast. The coefficients relating the glint contribution in the SWIR to that at VNIR wavelengths are then calculated over the entire GAA, ensuring that the derived image-specific coefficients optimally represent the average prevailing atmospheric conditions. It is further expected that the applied contrast-based minimization procedure proves robust in practical application, also in cases of high glint coverage.

This article contains all the necessary information to allow for GRCM's full implementation. To support the reader, the acronyms used in this manuscript are listed in Abbreviation at the end of the article.

## 2. Materials and Methods

### 2.1. Data Sources and Processing

#### 2.1.1. The Operational Land Imager

The OLI instrument flown onboard Landsat 8 provides operational imagery since 18 March 2013. The satellite orbits the Earth in a sun-synchronous, near-polar orbit with an inclination of 98.2 degrees at an altitude of 705 km crossing the equator at a Mean Local Time of 10:00 a.m. $\pm$ 15 min. OLI disposes of nine spectral bands ranging from the VIS to the SWIR at 12-bit radiometric resolution (Table 1).

**Table 1.** Operational Land Imager (OLI) spectral band characteristics. Adapted from the Landsat 8 Data Users Handbook [24].

| Spectral Band | Spectral Range | Spatial Resolution | Signal-to-Noise Ratio |
|---|---|---|---|
| B1, Coastal/Aerosol | 0.435–0.451 μm | 30 m | 238 |
| B2, Blue | 0.452–0.512 μm | 30 m | 364 |
| B3, Green | 0.533–0.590 μm | 30 m | 302 |
| B4, Red | 0.636–0.673 μm | 30 m | 227 |
| B5, NIR | 0.851–0.879 μm | 30 m | 204 |
| B6, SWIR-1 | 1.566–1.651 μm | 30 m | 265 |
| B7, SWIR-2 | 2.107–2.294 μm | 30 m | 334 |
| B8, Pan | 0.503–0.676 μm | 15 m | 149 |
| B9, Cirrus | 1.363–1.384 μm | 30 m | 165 |

Despite its name, and as evidenced by a large number of studies (see e.g., [25] and references therein), OLI is well suited for the observation of aquatic ecosystems due to its significantly improved radiometric resolution and signal-to-noise ratio as compared to its predecessors Thematic Mapper I and Enhanced Thematic Mapper Plus (ETM+) flown on Landsat 4/5 and Landsat 7, respectively [26].

### 2.1.2. OLI Level-1 to Level-2 Conversion

OLI Collection 2 Level 1 Terrain Precision (L1TP) data of Tier 1 have been used for the present work, offering consistent geo-registration within prescribed tolerances of <12 m radial root mean square error [27]. TOA reflectances have been calculated for each spectral band from the coefficients provided specified in the accompanying scene-specific metadata file according to the procedure described in [24] (Section 5.2):

$$\rho^{TOA} = \left( M_\rho \times Q_{cal} + A_\rho \right)/\cos\left(\theta_{SOL}\right), \tag{1}$$

where $\rho^{TOA}$ is the dimensionless planetary reflectance at TOA, $M_\rho$ is the reflectance multiplicative scaling factor, $A_\rho$ is the reflectance additive scaling factor, $Q_{cal}$ is the Level 1 pixel value in digital numbers, and $\cos\left(\theta_{SOL}\right)$ is the cosine of the local solar zenith angle.

For all investigated scenes, $M_\rho = 2.0 \times 10^{-5}$ [24]. The reflectance change per unit $Q_{cal}$ therefore amounts to:

$$\Delta\rho^{TOA} \text{ per unit } Q_{cal} = 2.0 \times 10^{-5}/\cos\left(\theta_{SOL}\right), \tag{2}$$

i.e., the reflectance resolution decreases with increasing solar zenith angle. For example, while OLI reflectance values are spaced $2.0 \times 10^{-5}$ for $\theta_{SOL} = 0°$, reflectance spacing increases to $4.0 \times 10^{-5}$ for $\theta_{SOL} = 60°$ and $7.7 \times 10^{-5}$ for $\theta_{SOL} = 75°$, respectively.

### 2.1.3. Areas of Interest

GRCM has been developed and tested using OLI subscenes from four areas of interest (AOIs) encompassing a wide range of environmental conditions (Table 2). For each AOI, two to five sample scenes have been processed (Table 3).

**Table 2.** Areas of interest (AOI) used in the development of the Glint Removal through Contrast Minimization scheme (GRCM). Each AOI covers an area of ca. $24 \times 36$ km$^2$.

| AOI Designation | Geographical Extension | Elevation above MSL | Description | Remarks |
|---|---|---|---|---|
| Brest [BRS] | 48.170–48.386°N 4.700–4.214°W | 0 m | Estuary and coastal waters of varying degrees of turbidity, frequent occurrence of swell from the open Atlantic. | AERONET [28] station "Brest_MF" within AOI. |
| Haifa Bay [HFA] | 32.750–32.966°N 34.714–35.100°E | 0 m | Inner Haifa Bay strongly impacted by anthropogenic activities (harbor), oligotrophic conditions offshore. | AERONET [28] station "Technion_Hai-fa_IL" within AOI. |
| Lake Con-stance East [LCE] | 47.450–47.666°N 9.270–9.750°E | 395 m | Large (536 km$^2$) and mostly oligotrophic lake in central Europe, intensively used for recreational purposes. | |
| Lake Puma Yumco [LPY] | 28.434–28.650°N 90.215–90.574°E | 5013 m | Large (280 km$^2$) oligotrophic lake on the Qinghai-Tibet Plateau, significantly reduced Rayleigh optical depth. | |

### 2.2. Morphological Aspects of Sun Glint

Sun glint is characterized in near-nadir high resolution imagery by specific reflectance patterns, exemplarily shown herein in Figure 1 for OLI sample scene BRS-3. Filament-like structures of low reflectance are observed within high reflectance areas, indicating locally lower surface roughness, e.g., around position "A" in both upper and lower inset. No sun glint is observed in the lee of the two little islands North of position "B". Ship wakes produce the typical glint pattern shown near position "C". Another typical glint pattern is produced by swell, characterized by increased reflectance parallel to the wave crests (e.g., swell propagating from the south-west to the north-east around position "D" in the lower

inset). These glint specific reflectance patterns differ significantly from the typically much smoother patterns caused by atmospheric turbidity or oceanic processes. Thin cirrus is observed near position "E".

**Table 3.** List of OLI sample scenes to which GRCM was applied.

| Sample Scene ID | Area of Interest | Date | Landsat Product ID | WRS2 Path/Row |
|---|---|---|---|---|
| BRS-1 | Brest | 13 May 2019 | LC08_L1TP_203026_20190513_20200828_02_T1 | 203/026 |
| BRS-2 | Brest | 4 April 2020 | LC08_L1TP_204026_20200404_20200822_02_T1 | 204/026 |
| BRS-3 | Brest | 23 June 2020 | LC08_L1TP_204026_20200623_20200823_02_T1 | 204/026 |
| BRS-4 | Brest | 10 August 2020 | LC08_L1TP_204026_20200810_20200918_02_T1 | 204/026 |
| HFA-1 | Haifa Bay | 9 January 2022 | LC08_L1TP_175037_20220109_20220114_02_T1 | 175/037 |
| HFA-2 | Haifa Bay | 15 April 2022 | LC08_L1TP_175037_20220415_20220420_02_T1 | 175/037 |
| HFA-3 | Haifa Bay | 10 May 2022 | LC08_L1TP_174037_20220510_20220518_02_T1 | 174/037 |
| HFA-4 | Haifa Bay | 26 May 2022 | LC08_L1TP_174037_20220526_20220602_02_T1 | 174/037 |
| HFA-5 | Haifa Bay | 11 June 2022 | LC08_L1TP_174037_20220611_20220617_02_T1 | 174/037 |
| LCE-1 | Lake Constance East | 22 July 2021 | LC08_L1TP_194027_20210722_20210729_02_T1 | 194/027 |
| LCE-2 | Lake Constance East | 1 June 2020 | LC08_L1TP_194027_20200601_20200824_02_T1 | 194/027 |
| LCE-3 | Lake Constance East | 19 July 2020 | LC08_L1TP_194027_20200719_20200911_02_T1 | 194/027 |
| LCE-4 | Lake Constance East | 20 August 2020 | LC08_L1TP_194027_20200820_20200905_02_T1 | 194/027 |
| LPY-1 | Lake Puma Yumco | 6 July 2018 | LC08_L1TP_138040_20180706_20200831_02_T1 | 138/040 |
| LPY-2 | Lake Puma Yumco | 8 September 2018 | LC08_L1TP_138040_20180908_20200831_02_T1 | 138/040 |

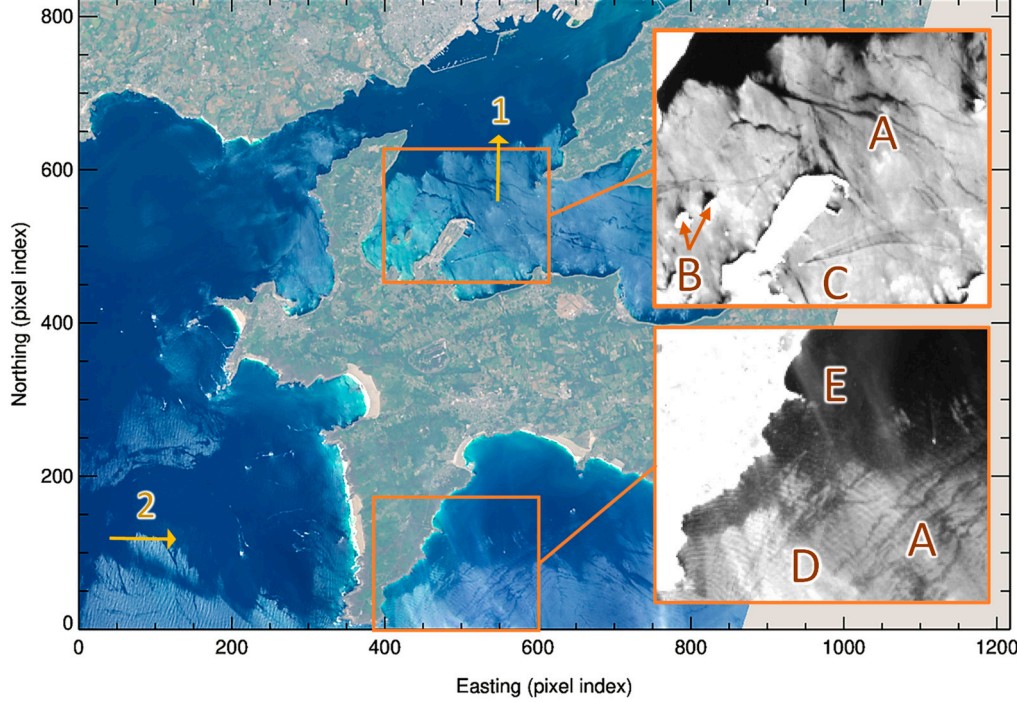

**Figure 1.** Contrast-enhanced near-true-color image derived from sample scene BRS-3, using OLI channels B2, B3, and B4. The two yellow lines labeled "1" and "2" indicate the transects analyzed in Figure 2. The insets on the right show the shortwave infrared (SWIR) reflectance in channel B7 for two subareas with the letters "A" to "E" pointing to characteristic glint reflectance patterns. See text for further explanation.

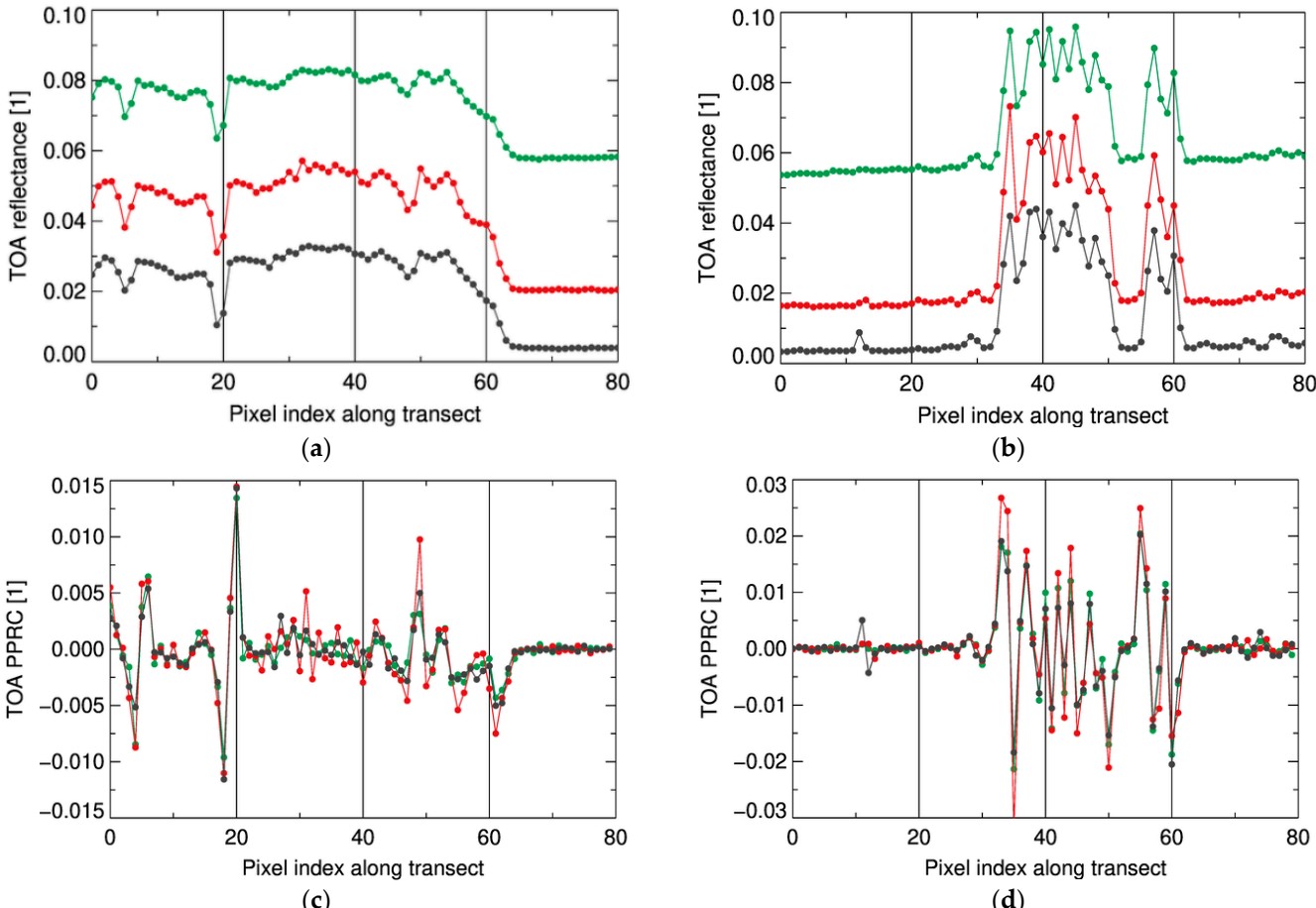

**Figure 2.** (**a**) Top-of-atmosphere (TOA) reflectance along Transect "1" (pixel indices 550/560 to 550/640) indicated in Figure 1; (**c**) Corresponding pixel-to-pixel reflectance contrast for OLI channels B3 (green), B5 (red), and B7 (dark grey); (**b,d**) Equivalent to (**a,c**), but for Transect "2" (pixel indices 40/120 to 120/120). Unit [1] on the y-axes represents the SI unit for dimensionless quantities.

Aside the typical larger features spreading over several pixels, sun glint is also characterized by significantly increased local reflectance contrast at TOA as compared to glint-free areas. Figure 2a shows the TOA reflectance for OLI channels B3, B5, and B7 along Transect "1" indicated in Figure 1, extending from high glint ($\rho^{TOA}$ ($B7$) $\approx 0.03$) into glint-free areas starting at pixel index #64 ($\rho^{TOA}$ ($B7$) $< 0.005$). While the corresponding pixel-to-pixel reflectance contrast (PPRC) depicted in Figure 2c is small in the glint-free areas |$PPRC$| $< 0.001$), it strongly varies in GAAs reaching values of |$PPRC$| $> 0.01$. Similar observations are made in Figure 2b,d for Transect "2", where the glint signal is further enhanced by the presence of swell between ca. pixel indices #35 to #50 and again #55 to #60.

The following conclusions can be drawn from the visualization of the sun glint's morphology. First, OLI's spatial resolution of 30 m is sufficient to resolve reflectance patterns caused by sun glint. Second, sun glint in OLI imagery is characterized by enhanced local reflectance contrasts as compared to neighboring glint-free areas. Third, there is no obvious spatial shift between the different OLI channels, i.e., all OLI channels observe the glint patterns in a very similar way (see for example the swell-induced local reflectance maxima in Transect "2").

*2.3. A Sun Glint Mask Derived from the Local Reflectance Contrast in the SWIR*

2.3.1. Identifying Sun Glint Affected Pixels and Areas

The identification of glint affected pixels and areas presented herein is based on the maximum reflectance contrast (*MRC*), defined as a local contrast measure within a $3 \times 3$ pixel area centered at position $(i, j)$ by:

$$MRC \equiv mrc_{i,j} := \max_{\substack{i-1 \le k \le i+1 \\ j-1 \le l \le j+1}} \left( \rho_{i,j}^{TOA}(B_7) - \rho_{k,l}^{TOA}(B_7) \right). \tag{3}$$

*MRC* adopts values greater or equal zero, the latter if $\rho_{i,j}^{TOA}(B_7)$ takes on the minimum reflectance within the corresponding $3 \times 3$ pixel area. Otherwise, *MRC* represents the reflectance contrast between $\rho_{i,j}^{TOA}(B_7)$ and its darkest neighboring pixel.

In a second step, the mask $MSK_{PGP}$ of potentially glinted pixels (subscript "*PGP*" in equations) is created by applying a threshold $THR_{PGP}$:

$$MSK_{PGP} \equiv msk\_pgp_{i,j} := \begin{cases} 1 \text{ if } mrc_{i,j} > THR_{PGP}, \\ 0 \text{ otherwise,} \end{cases} \tag{4}$$

where $THR_{PGP}$ needs to be chosen such that the contrast produced by instrumental noise or environmental processes such as local variations of the aerosol reflectance are not mistaken for sun glint. $MSK_{PGP}$ then identifies all pixels where the local contrast is strong enough to assume the presence of glint. (See Section 2.3.2. on how $THR_{PGP}$ as well as the other thresholds introduced in this Section 2.3.1. are practically determined.)

Environmental conditions leading to sun glint are typically spreading over areas significantly larger than that represented by an individual OLI pixel. Sun glint affected pixels rarely come "alone", but rather congregate in glint prone areas. This reasoning leads to the following criterion to remove pixels that likely have erroneously been classified as being potentially glinted:

$$MSK_{GAP} \equiv msk\_gap_{i,j} := \begin{cases} 1 \text{ if } msk\_pgp_{i,j} = 1 \ \wedge \ \underset{M \times M}{\text{mean}} \left( msk\_pgp_{i,j} \right) > THR_{GAP}, \\ 0 \text{ otherwise,} \end{cases} \tag{5}$$

i.e., a pixel $msk\_pgp_{i,j}$ classified as potentially glinted according to Equation (4) is considered an actually glint affected pixel (subscript "*GAP*" in equations) $msk\_gap_{i,j}$ if the relative coverage by potentially glinted pixels $msk\_pgp_{i,j}$ within the $M \times M$ window centered at position $(i, j)$ exceeds threshold $THR_{GAP}$.

Not all glinted pixels are characterized by strong contrast against their neighbors, for example, if the latter are similarly glinted, meaning that Equation (5) does not identify glint affected pixels in their entirety. A pixel $\rho_{i,j}^{TOA}$ is therefore considered to be located in a GAA if the relative coverage of glint affected pixels within the $N \times N$ window centered at position $(i, j)$ exceeds threshold $THR_{GAA}$:

$$MSK_{GAA} \equiv msk_{gaa_{i,j}} := \begin{cases} 1 \text{ if } \underset{N \times N}{\text{mean}} \left( msk\_gap_{i,j} \right) > THR_{GAA}, \\ 0 \text{ otherwise.} \end{cases} \tag{6}$$

2.3.2. Determining the Model Parameters for the Sun Glint Mask

To identify the GAA by applying the method devised in Section 2.3.1, thresholds $THR_{PGP}$, $THR_{GAP}$, $THR_{GAA}$, as well as the window widths $M$ and $N$ need to be determined.

For the threshold contrast $THR_{PGP}$, i.e., the *MRC* value above which a pixel is considered as potentially glinted, this has been done as follows:

- Eight low cloud cover OLI scenes were preselected, encompassing an otherwise wide range of environmental conditions.
- Within each scene, one cloud-free area of negligible glint occurrence was determined from visual inspection.

- For each such cloud and glint-free area, the 99th percentile ($P_{99}$) of *MRC* in channel B7 was determined.

Figure 3 presents the *MRC* $P_{99}$ values for the eight cloud and glint-free areas as a function of the solar zenith, together with a fit representing $THR_{PGP}$ by:

$$THR_{PGP}(\theta_{sol}) = 0.0005 / \cos(0.95 \times \theta_{sol}). \tag{7}$$

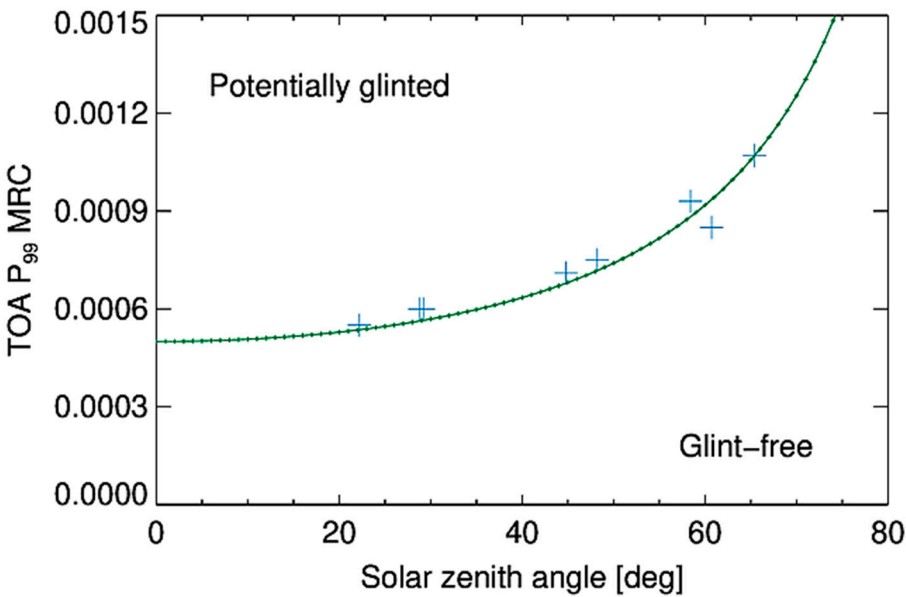

**Figure 3.** $P_{99}$ of the maximum reflectance contrast (MRC) in OLI channel B7 for eight glint-free areas as a function of the solar zenith angle. The fit represents threshold $THR_{PGP}$ in Equation (4) above which a pixel is considered potentially sun glint affected.

The dependence of $THR_{PGP}$ on the solar zenith angle is due to the fact that the reflectance change per unit Level-1 pixel value increases with increasing solar zenith according to Equation (2).

The window widths $M$ and $N$ as well as the thresholds $THR_{GAP}$ and $THR_{GAA}$ have been chosen as:

$$\begin{aligned} M &= 5, \\ THR_{GAP} &= (1+4)/M^2 = 0.2, \\ N &= 3, \\ THR_{GAA} &= 1/N^2 \cong 0.11, \end{aligned} \tag{8}$$

based on the following reasoning:

- A potentially glinted pixel at position ($i$, $j$) is considered glint affected if at least four further pixels within a $5 \times 5$ window centered at ($i$, $j$) are also potentially glinted, hence, $THR_{GAP} = (1+4)/25 = 0.2$.
- A pixel (whether glint affected or not) at position ($i$, $j$) is considered part of a GAA if at least one glint affected pixel is located within a $3 \times 3$ window centered at ($i$, $j$), hence, $THR_{GAA} = 1/9 \cong 0.11$.

These thresholds represent pragmatic solutions based on practical experience and have been used for the processing of all scenes presented herein. They likely need to be chosen differently for sensors other than OLI to provide reliable results.

## 2.4. Contrast-Based Estimation of the Sun Glint at TOA

Under cloud-free conditions, the top-of-atmosphere radiance $L^{TOA}$ above water can be decomposed as follows [17]:

$$L^{TOA} = L^{TOA}_{RAY} + L^{TOA}_{AER} + L^{TOA}_{AR} + T \times L^{0+}_{SUG} + t \times L^{0+}_{WCP} + t \times L^{0+}_{WAT}, \tag{9}$$

where $L_{RAY}^{TOA}$ and $L_{AER}^{TOA}$ designate the contributions of Rayleigh (including sky glint) and aerosol scattering to the TOA radiance, $L_{AR}^{TOA}$ is a coupling term accounting for the interaction between aerosol and Rayleigh scattering, $L_{SUG}^{0+}$ designates the sun glint just above the sea surface, attenuated on its way to the sensor by the direct atmospheric transmittance $T$, whereas $L_{WCP}^{0+}$ and $L_{WAT}^{0+}$ represent the contributions from white caps and the water leaving radiance just above the sea surface, subsequently attenuated by the diffuse atmospheric transmittance $t$. Note that all parameters in Equation (9) depend on wavelength and observation geometry, indicated further only if required to enhance comprehensibility.

In the SWIR, the terms $L_{RAY}^{TOA}$ and $L_{AR}^{0+}$ are negligible since the atmospheric Rayleigh optical depth is very low [29]. The same is true for $L_{WCP}^{0+}$ and $L_{WAT}^{0+}$ due to the very high absorption of pure water [22,30] such that Equation (9) simplifies to:

$$L^{TOA}(\lambda_{SWIR}) = L_{AER}^{TOA}(\lambda_{SWIR}) + T(\lambda_{SWIR}) \times L_{SUG}^{0+}(\lambda_{SWIR}). \qquad (10)$$

Introducing:

$$L_{SUG}^{TOA}(\lambda_{SWIR}) = T(\lambda_{SWIR}) \times L_{SUG}^{0+}(\lambda_{SWIR}), \qquad (11)$$

Equation (10) can be rearranged to:

$$L_{SUG}^{TOA}(\lambda_{SWIR}) = L^{TOA}(\lambda_{SWIR}) - L_{AER}^{TOA}(\lambda_{SWIR}), \qquad (12)$$

i.e., once the aerosol contribution to the TOA radiance in the SWIR is known, the sun glint contribution at TOA can be assessed. Dividing both sides of Equation (12) by the factor $E_d^{TOA}/\pi$, where $E_d^{TOA}$ is the downwelling irradiance at TOA, one obtains the equivalent formulation in reflectance units $\rho$:

$$\rho_{SUG}^{TOA}(\lambda_{SWIR}) = \rho^{TOA}(\lambda_{SWIR}) - \rho_{AER}^{TOA}(\lambda_{SWIR}). \qquad (13)$$

The spectral dependence of the sun glint reflectance at TOA can be expressed as

$$\rho_{SUG}^{TOA}(\lambda) = c(\lambda_{SWIR}, \lambda) \times \rho_{SUG}^{TOA}(\lambda_{SWIR}), \qquad (14)$$

where $c(\lambda_{SWIR}, \lambda)$ is a scalar factor quantifying the glint reflectance at target wavelength $\lambda$ relative to the reference wavelength $\lambda_{SWIR}$, further on referred to in the text as TOA Spectral Glint Conversion (TSGC). In principle, TSGC varies across a satellite scene as it depends on both observation geometry and atmospheric transmittance (see Section 2.5) but is considered constant for the limited size ($24 \times 36$ km$^2$) of the AOIs considered herein. Obviously, such simplification does not apply to inland water surfaces at differing altitudes within a single scene. In such case, water bodies at different altitudes need either to be processed independently, or a Rayleigh correction needs to be applied prior to glint correction.

As sun glint usually produces more contrasted patterns at TOA in cloud-free areas than do other atmospheric or oceanic processes (see Section 2.2), removing the sun glint results in a contrast reduction; TSGC is therefore chosen correctly if the total contrast within the sun glint corrected image $\rho_{COR}^{TOA}(\lambda)$ at the target wavelength, defined by:

$$\rho_{COR}^{TOA}(\lambda) = \rho^{TOA}(\lambda) - c(\lambda_{SWIR}, \lambda) \times \rho_{SUG}^{TOA}(\lambda_{SWIR}), \qquad (15)$$

adopts a minimum:

$$c(\lambda_{SWIR}, \lambda) := \min_{c\prime \in [0, \, C\prime]} f_c \left( \rho^{TOA}(\lambda) - c\prime \times \rho_{SUG}^{TOA}(\lambda_{SWIR}) \right), \qquad (16)$$

where $f_c$ is a suitable measure of contrast and $[0, C\prime]$ represents a sensible range of values for TSGC. The practical implementation of GRCM is described in Section 3.

### 2.5. Sun Glint Based Estimation of the Spectral Atmospheric Transmittance

Introducing the atmospheric direct transmission $T(\lambda, \mu)$, where $\mu$ indicates the cosine of the solar ($\mu_S$) and observational ($\mu_O$) zenith angle, respectively, Equation (14) can be expressed in terms of sun glint reflectance just above the water surface $\rho^{0+}_{SUG}$:

$$\rho^{0+}_{SUG}(\lambda) \times T(\lambda, \mu_S) \times T(\lambda, \mu_O) =$$
$$c(\lambda_{SWIR}, \lambda) \times \rho^{0+}_{SUG}(\lambda_{SWIR}) \times T(\lambda_{SWIR}, \mu_S) \times T(\lambda_{SWIR}, \mu_O). \tag{17}$$

The glint reflectance just above the water surface $\rho^{0+}_{SUG}$ is determined by the Bidirectional Reflectance Distribution Function (BRDF) of the rough water surface and varies as a function of wavelength due to the spectral dependence of the refractive index of water. Introducing the spectrally normalized BRDF $\varepsilon$ by:

$$\varepsilon(\lambda_{REF}, \lambda) = BRDF(\lambda) / BRDF(\lambda_{REF}), \tag{18}$$

and setting $\lambda_{REF} = \lambda_{SWIR}$, it follows that:

$$\rho^{0+}_{SUG}(\lambda) = \varepsilon(\lambda_{SWIR}, \lambda) \times \rho^{0+}_{SUG}(\lambda_{SWIR}). \tag{19}$$

Radiative transfer calculations have been used by [13] to determine the spectral dependence of $\varepsilon$. Setting the reference wavelength to 2190 nm, i.e., the central wavelength of OLI's channel B7, it increases from 1.0 at 2190 nm to ~1.25 at 500 nm. Introducing Equation (19) into Equation (17) gives:

$$c(\lambda_{SWIR}, \lambda) = \varepsilon(\lambda_{SWIR}, \lambda) \times \frac{T(\lambda, \mu_S) \times T(\lambda, \mu_O)}{T(\lambda_{SWIR}, \mu_S) \times T(\lambda_{SWIR}, \mu_O)}. \tag{20}$$

Knowing both parameters TSGC and $\varepsilon$ allows to deduce information on the spectral dependence of the atmospheric transmission (and hence optical depth) against a reference wavelength, as already stated by [31] and further discussed in Section 4.4.

Summarizing, GRCM consists of three principal steps. First, the maximum reflectance contrast metric is applied to SWIR reflectances at TOA to identify the entire glint affected area. A contrast minimization scheme is then invoked to relate the sun glint contribution in the SWIR to the corresponding contribution in the VNIR. Subtracting the such-derived glint contribution finally provides glint corrected TOA reflectances in the VNIR.

## 3. Results

### 3.1. Overview of the GRCM Implementation

The flow chart in Figure 4 provides a top-level description of the practical implementation of GRCM. The individual processing steps are indicated by the letters [A] to [H], outlined in more detail in the following subsections, and applied to sample scene BRS-3 for illustration purposes.

### 3.2. Preparing GRCM

3.2.1. [A] Correcting the TOA Reflectance for Absorption by Atmospheric Gases

The atmospheric absorption due to $CO_2$, $H_2O$, $O_2$, and $O_3$ cannot be neglected for certain OLI channels. Its impact on TOA reflectance can be reasonably well determined by assuming gaseous absorption taking place above the top of the scattering atmosphere:

$$\rho^* = \rho^{TOA} / \left( T^{du}_{CO_2} \times T^{du}_{O_2} \times T^{du}_{O_3} \times T^{du}_{H_2O} \right), \tag{21}$$

where $T^{du}$ is the double path (downward and upward) gaseous transmittance, and $\rho^*$ designates the TOA reflectance corrected for atmospheric gaseous absorption.

In the present work, atmospheric gaseous transmittance is calculated using the SMAC (Simplified Method for Atmospheric Correction) approach [32]. The required band-specific coefficients for the OLI instrument are obtained from [33], while the atmospheric parameters

(total column ozone, total column water vapor, mean sea level atmospheric pressure) are taken from ERA5 hourly data on single levels [34].

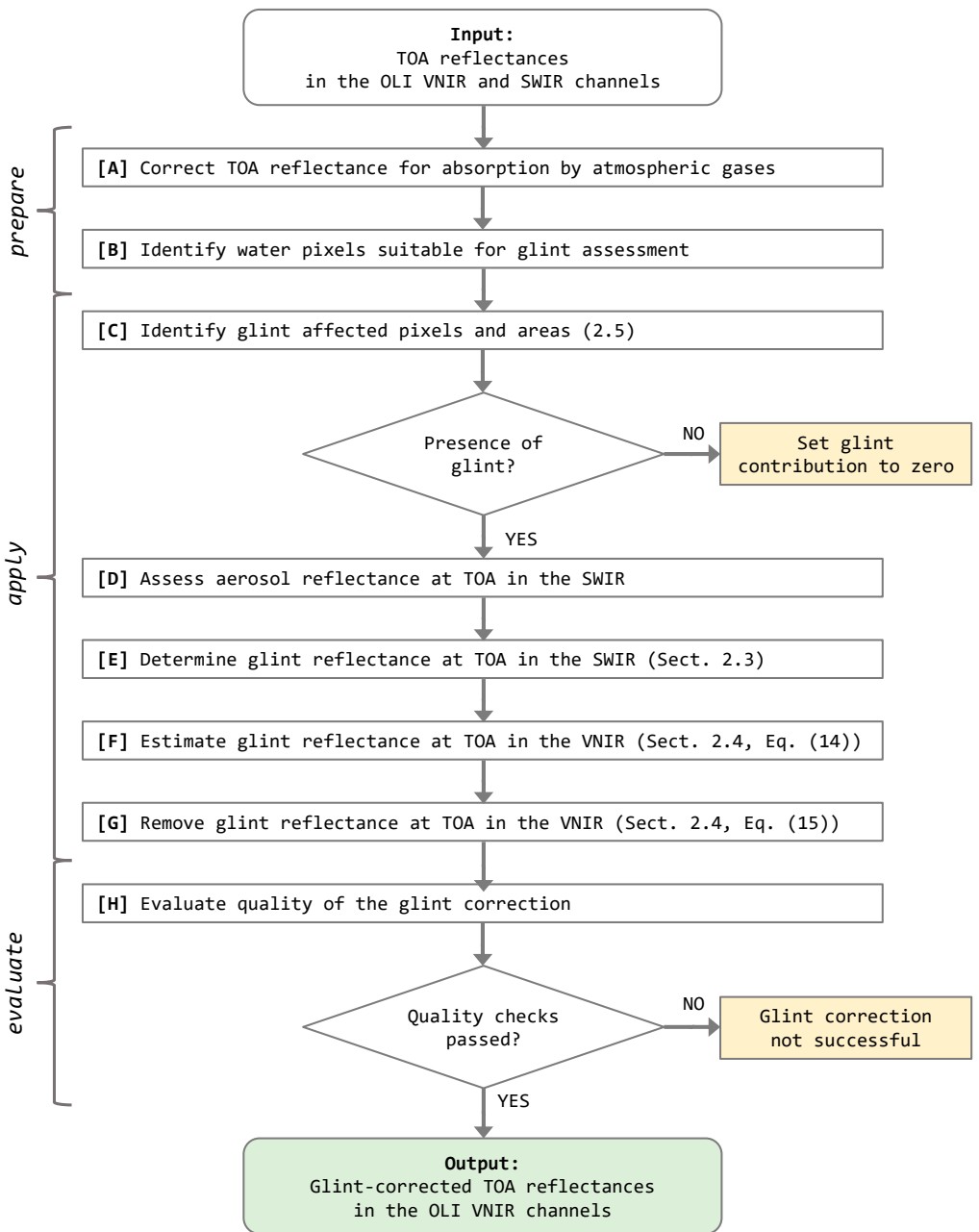

**Figure 4.** Flow chart of the GRCM sun glint correction scheme.

### 3.2.2. [B] Identifying Water Pixels Suitable for Glint Assessment

There are a number of environmental conditions with potentially negative impact on GRCM, most importantly the presence of clouds and cloud shadows, but also the occurrence of white caps, bottom-up effects in shallow waters, bright objects on the water surface, etc. While some of these conditions do not significantly affect reflectances in the SWIR due the strong in-water absorption of light at these wavelengths, they may "confuse" contrast minimization in the VNIR. The concerned pixels therefore need to be identified and possibly excluded from further processing.

In the context of the present study, visual inspection using all OLI channels has been applied to avoid the presence of clouds and cloud shadows in the investigated areas of interest. The corresponding masks $MSK_{CLD}$ and $MSK_{SHD}$ are therefore set to zero:

$$MSK_{CLD} \equiv msk\_cld_{i,j} := 0, \qquad (22)$$

$$MSK_{SHD} \equiv msk\_shd_{i,j} := 0. \qquad (23)$$

A normalized difference water index (NDWI) defined by:

$$\text{NDWI} = (\rho^*(B7) - \rho^*(B3)) \,/\, (\rho^*(B7) + \rho^*(B3)), \qquad (24)$$

is applied to identify water pixels. Similar indices have proven successful in identifying water surfaces in satellite imagery [35]. The NDWI defined through Equation (24) usually adopts values below $-0.5$ at TOA above water, while being positive above land surfaces. In the presence of sun glint or increased atmospheric aerosol, the NDWI above water increases but remains negative. Therefore, the water mask $MSK_{WAT}$ is defined herein as:

$$MSK_{WAT} \equiv msk\_wat_{i,j} := \begin{cases} 1 \text{ if } ndwi_{i,j} < THR_{WAT}, \\ \quad 0 \text{ otherwise}, \end{cases} \qquad (25)$$
$$\text{with } THR_{WAT} = -0.2.$$

Pixels classified as water according to Equation (25) but affected by bright objects on the water such as vessels, offshore platforms, etc. need to be excluded as the corresponding sharp reflectance contrasts in the SWIR may erroneously be identified as sun glint. The bright pixel mask $MSK_{BGT}$ applies a simple empirical threshold approach to identify such pixels:

$$MSK_{BGT} \equiv msk\_bgt_{i,j} := \begin{cases} 1 \text{ if mean } \left( \left\{ \rho^*_{i,j}(B3), \, \rho^*_{i,j}(B5), \rho^*_{i,j}(B7) \right\} \right) < THR_{BGT}, \\ \quad 0 \text{ otherwise}, \end{cases} \qquad (26)$$
$$\text{with } THR_{BGT} = 0.08.$$

Water pixels close to the shore or near bright objects on the water need to be excluded to avoid the presence of mixed pixels potentially producing enhanced contrasts. To this aim, an inward (i.e., negative) five-pixel buffer $MSK_{BUF}$ has been applied to the dynamic water mask $MSK_{WAT}$.

The above defined masks are combined to identify pixels $MSK_{GOOD}$ suitable for the application of GRCM ($\wedge$: logical AND, $\neg$: logical NOT):

$$MSK_{GOOD} \equiv msk\_good_{i,j} :=$$
$$MSK_{WAT} \wedge (\neg\, MSK_{CLD}) \wedge (\neg\, MSK_{SHD}) \wedge (\neg\, MSK_{BGT}) \wedge (\neg\, MSK_{BUF}). \qquad (27)$$

### 3.3. Applying GRCM

#### 3.3.1. [C] Identifying Glint Affected Pixels and Areas

The stepwise identification of the GAA as described in Section 2.3 is depicted in Figure 5 at the example of sample scene BRS-3. Of all pixels classified as water according to Equation (25), 87.9% are assessed suitable ("good") according to Equation (27) for use in GRCM. MRC in OLI channel B7 calculated from Equation (3) is used to identify the glint-affected pixels according to Equation (5), amounting to 56.7% of all suitable pixels, finally resulting in a GAA according to Equation (6) with a coverage of 72.1%.

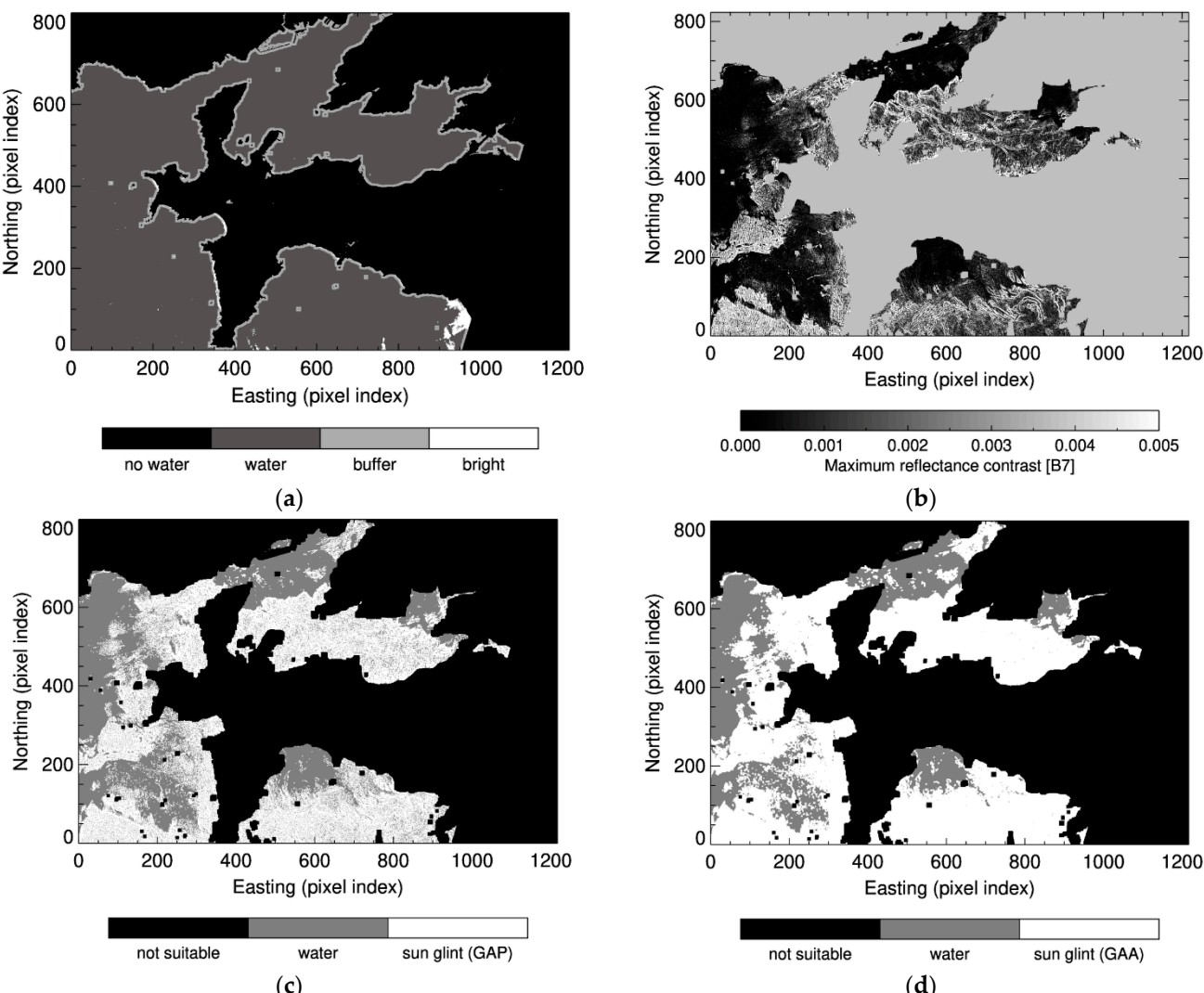

**Figure 5.** Determining the extension of the glint affected area (GAA) at the example of sample scene BRS-3. (**a**) Identification of pixels suitable for use in GRCM; (**b**) TOA MRC for OLI channel B7; (**c**) Glint affected pixels (GAP); (**d**) GAA. See text for further explanation.

If no sun glint is detected in an image, the glint reflectance at TOA is set to zero and glint processing stops.

### 3.3.2. [D] Assessing the Aerosol Reflectance at TOA in the SWIR

In order to calculate the sun glint contribution $\rho^*_{SUG}$ to the total TOA reflectance $\rho^*$ in the SWIR, the corresponding aerosol contribution $\rho^*_{AER}$ needs to be assessed. This is done by calculating the $P_1$ percentile in OLI channel B7 of all pixels considered not being glint affected:

$$\rho^*_{AER}(B7) = P_1\{\rho^*(B7) \mid MSK_{GOOD} \wedge (\neg\, MSK_{GAP})\}. \tag{28}$$

The SWIR aerosol reflectance defined through Equation (28) is applied to the entire AOI; it is not attempted to assess the spatial variability of the aerosol reflectance within the scene. For sample scene BRS-3, $\rho^*_{AER}(B7)$ amounts to 0.0031, indicating rather low atmospheric turbidity.

### 3.3.3. [E] Determining the Glint Reflectance at TOA in the SWIR

Having determined $\rho_{AER}^*(B7)$, the TOA glint reflectance in the SWIR is obtained by applying Equation (13):

$$\rho_{SUG}^*(B7) := \begin{cases} \rho^*(B7) - \rho_{AER}^*(B7) \text{ if } \rho^*(B7) \geq \rho_{AER}^*(B7), \\ 0 \text{ otherwise.} \end{cases} \tag{29}$$

### 3.3.4. [F] Estimating the Glint Reflectance at TOA in the VNIR

Once the glint reflectance at TOA in the SWIR $\rho_{SUG}^*(B7)$ is known, it can be used to estimate TSGC as described in Section 2.4:

$$c(B7, B_n) := \min_{c\prime \in [0, 1.5]} (f_c [\rho^*(B_n) - c\prime \times \rho_{SUG}^*(B7)]), \tag{30}$$

where $B_n$ is the OLI channel to be sun glint corrected, $f_c$ is chosen as the average MRC over the entire GAA, further on referred to as AMRC, and $c\prime$ is varied within the range [0.0, 1.5] which has shown sufficient to cover the combined spectral dependence on sun glint and atmospheric transmission for all OLI channels.

Figure 6a demonstrates the contrast minimization procedure for sample scene BRS-3 at the example of OLI channel B3 (0.562 µm): subtracting increasing portions of the SWIR sun glint leads to decreasing AMRC values until a minimum is reached at $c\prime(B7, B3) = 0.96$, beyond which overcorrection sets in. The maximum AMRC reduction ($\Delta AMRC$) depends on sun glint intensity and amounts to $\Delta AMRC = 0.00157$ in OLI channel B3.

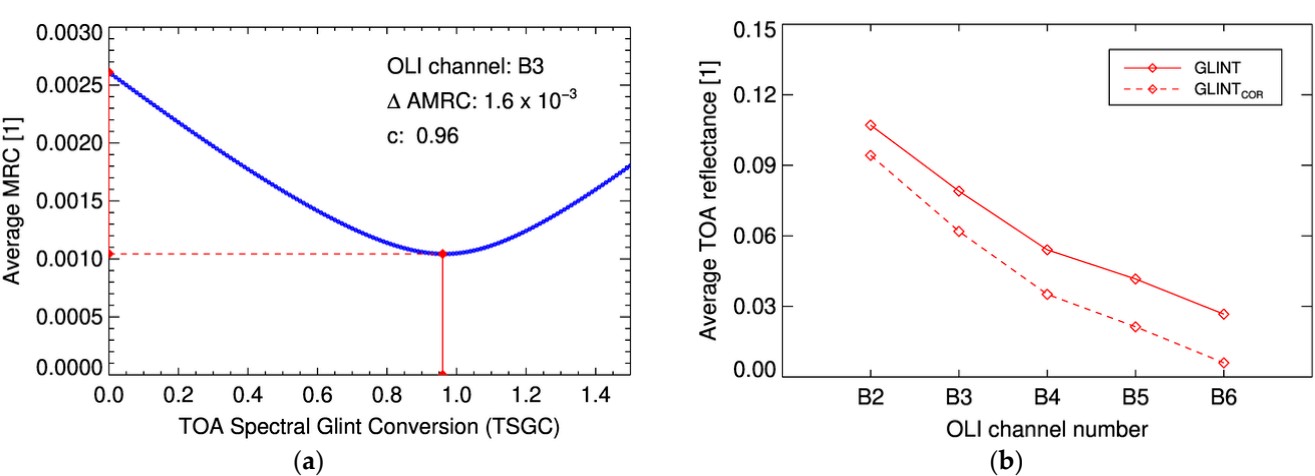

**Figure 6.** (**a**) Graphical representation of the minimization procedure to determine the TOA Spectral Glint Conversion (TSGC) at the example of OLI channel B3 for sample scene BRS-3; (**b**) Mean spectral TOA reflectance before and after glint correction. See text for further explanation.

### 3.3.5. [G] Removing the Glint Reflectance at TOA in the VNIR

After TSGC has been determined, the TOA reflectance in OLI channel $B_n$ can be corrected for the sun glint contribution following Equation (15):

$$\rho_{COR}^*(B_n) = \rho^*(B_n) - c(B7, B_n) \times \rho_{SUG}^*(B7). \tag{31}$$

This correction is applied to all pixels classified as water according to Equation (25).

The impact of the sun glint correction is demonstrated in Figure 6b, showing the mean TOA reflectance for OLI channels B2 to B6 before (solid line) and after (dashed line) sun glint correction. The absolute amount of the correction increases with increasing wavelength from 0.0129 for OLI channel B2 to 0.0203 for OLI channel B5. This is caused by increasing atmospheric transmission towards longer wavelengths overcompensating the increase of the glint reflectance at the water surface towards shorter wavelengths.

### 3.4. Evaluating GRCM

#### 3.4.1. [H-1] Qualitative Evaluation

The success of the glint correction procedure for a particular image can be assessed in a qualitative way through visual inspection: if previously obscured water-leaving reflectance patterns become visible after correction, a significant portion of the sun glint must have been removed, allowing for a better general understanding of the in-water biogeochemical processes or bathymetric features, especially by experts with in-depth knowledge of the investigated area. If glint patterns persist, or if artefacts are created, then the correction was obviously less successful.

This is illustrated by Figure 7 showing the TOA reflectance in OLI channel B3 before (a) and after (c) sun glint correction for sample scene BRS-3. Sun glint has been successfully removed over the whole scene. Details become visible when zooming into a $6 \times 6$ km$^2$ subarea: the glint-corrected TOA reflectance (d) reveals highly dynamic conditions in the north-eastern estuaries which are partly or completely outshone in the corresponding non-corrected product (b). While the correction has removed most of the glint, it has also introduced some smaller artefacts, especially along the filament-like structures close to the left image border. Two reasons could be responsible for this: either the correction coefficients derived from the entire image do not perfectly fit the local conditions, and/or there is a spatial misalignment between the concerned channels B3 and B7. In both cases, artefacts will be most pronounced where there are strong local contrasts.

The outcome of GRCM for three further sample scenes HFA-5, LCE-2, and LPY-1 (Table 3) are shown and discussed in Appendix A.

#### 3.4.2. [H-2] Quantitative Evaluation

To assess the success of the glint correction in a quantitative manner is more challenging. Several metrics have been tentatively defined towards this aim.

Assuming sun glint being only loosely correlated with the atmospheric and water leaving contributions at TOA, the average reflectance difference $\Delta$REF between glint affected pixels and neighboring glint-free areas at distances $\leq 5$ pixels should be small after correction. Values of $|\Delta REF| > \sim 0.001$ may hint at low performance of GRCM for a particular scene and/or channel. Indicating the general success of the glint correction for sample scene BRS-3, this condition is met for all OLI channels except channel B5 (Figure 8) slightly exceeding the threshold ($\Delta REF \cong 0.0012$), possibly resulting from the rather low SNR for this channel.

The background TOA aerosol reflectance $\rho_{AER}^*$ in the SWIR is required to isolate the glint reflectance. Values of $\rho_{AER}^*(B7) > 0.005$ may hint to either a high atmospheric aerosol load or the occurrence of non-negligible sun glint outside the GAA, the latter with negative consequences on GRCM performance.

GRCM relies on the occurrence of exploitable reflectance contrasts. In case $\Delta AMRC$ (see Section 3.3.4) adopts values of $\Delta AMRC < 2 \times 10^{-4}$, the glint signal may be insufficient to allow for an accurate determination of TSGC.

The thresholds to assess glint correction quality given in this section are based on practical experience. Above-threshold quality metrics do not necessarily indicate failure of GRCM but should trigger a critical review of results. Further refined quality evaluation schemes will have to be devised for the automated processing of larger image quantities.

A comparison of glint corrected reflectances against in-situ measurements taken just above the sea surface has not been attempted as this would involve an additional atmospheric correction step (remember that GRCM removes the glint contribution to the TOA reflectance) which is deemed beyond the scope of the present work.

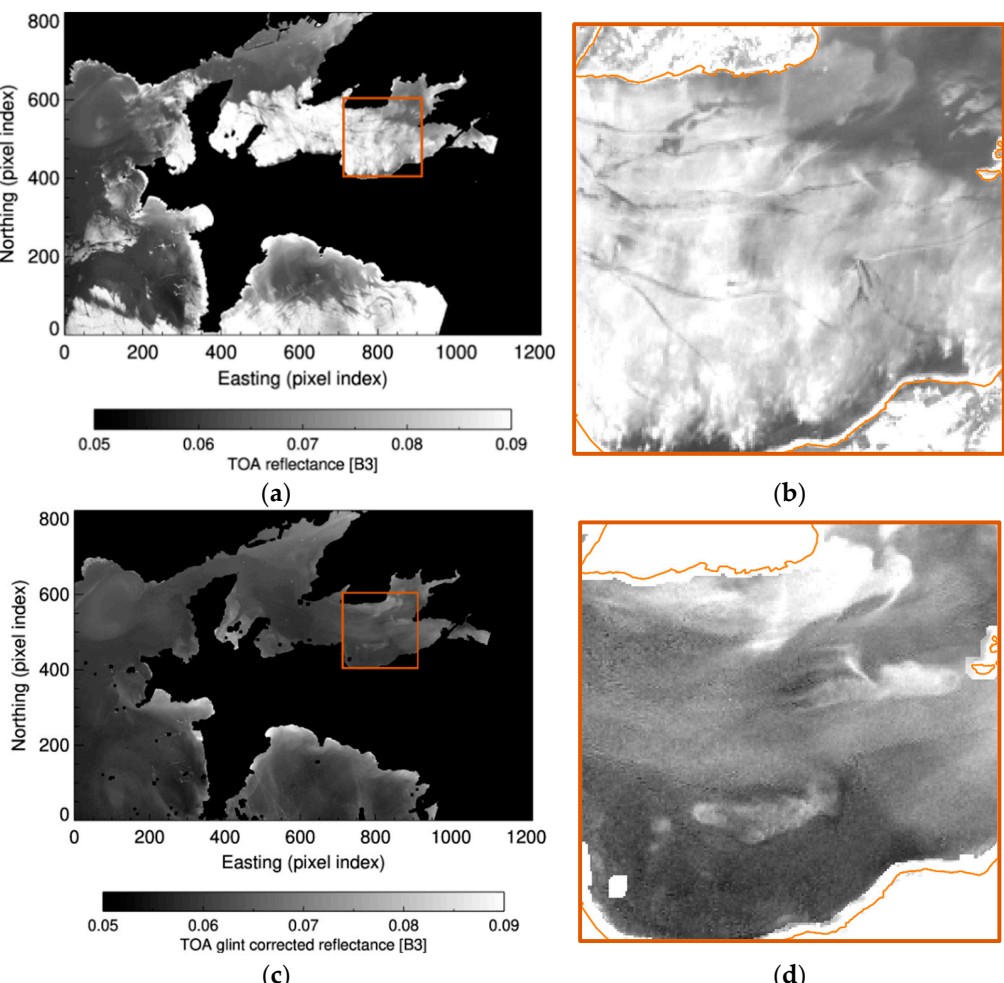

**Figure 7.** TOA reflectance in OLI channel B3 for sample scene BRS-3 before (**a**) and after (**c**) glint correction; (**b**) Enlarged detail covering an area of 6 × 6 km² before correction at identical reflectance grey scale [0.05–0.09]; (**d**) Enlarged detail after correction using an optimized grey scale [0.055–0.07]. See text for further explanation.

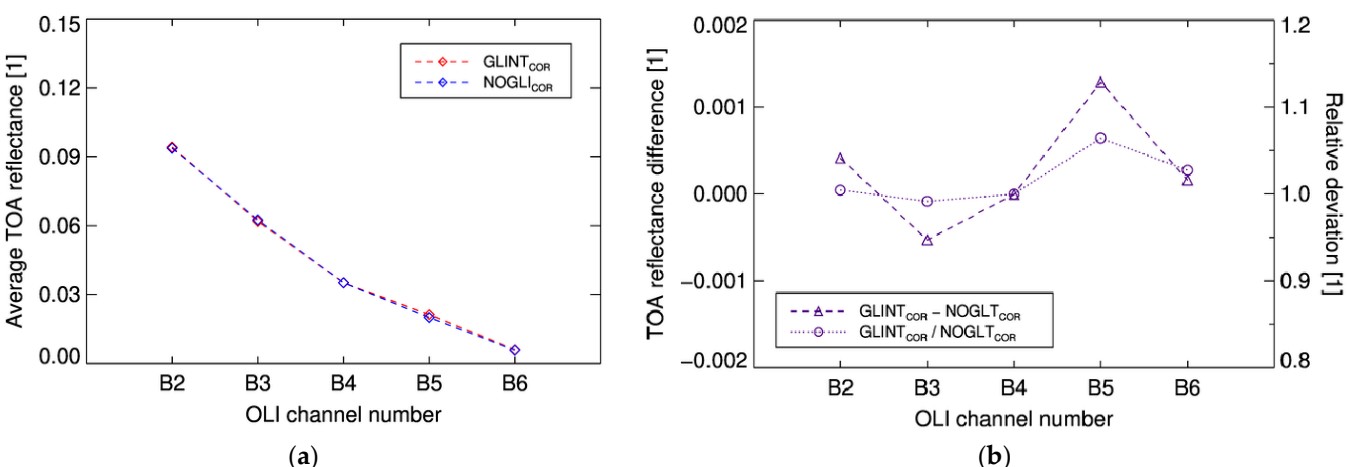

**Figure 8.** (**a**) Average TOA reflectance for sample scene BRS-3 after GRCM application in OLI channels B2 to B6 for glint-affected pixels (red) and neighboring glint-free pixels (blue); (**b**) Absolute (dashed, left axis) and relative (dotted, right axis) differences between glint-affected and neighboring glint-free pixels.

## 4. Discussion

### 4.1. Requirements on the Observing Imager

A number of requirements must be met by the observing imager to allow for the application of GRCM:

- The imager must be able to resolve morphological fine structures typical of sun glint, requiring a spatial resolution of $\leq$ca. 50 m.
- The imager must dispose of at least one channel in the SWIR, preferably at wavelengths $\geq 2.0$ μm to avoid sub-surface contributions.
- All applied spectral channels need to provide approximately identical representations of the observed water surface in terms of observation time, spatial resolution, and image registration.

As of 2022, OLI onboard Landsat 8 as well as OLI-2 onboard Landsat 9 appear to be the operational instruments best matching the above conditions. The hyperspectral EnMap mission [36] successfully launched in April 2022 will likely also meet the requirements for application of the proposed method, offering 30 m spatial resolution for the spectral range from 420 to 2450 nm with only a short temporal delay between VNIR and SWIR observations of 88 ms.

This is not the case for the MSI instrument onboard Sentinel-2 A/B where VIS, NIR, SWIR channels operate at different spatial resolutions (10 m, 20 m, 60 m) and, even more importantly, the different channels observe the same surface area with temporal delays of up to several seconds [37]; the rapidly changing sun glint patterns are therefore not identically represented in the different MSI channels. These issues might partly be overcome by applying averaging procedures at the price of reduced spatial resolution, but this has not been further investigated in this work.

### 4.2. Practical Application

GRCM has been applied to 15 OLI scenes from the years 2018 to 2022 experiencing GAA coverage between 5% and 100%. Preliminary conclusions on strengths and weaknesses of the current implementation as well as suggestions for further improvements can be drawn from the retrieval statistics and quality metrics shown in Table 4.

The average TOA reflectance after correction is very similar ($|\Delta REF(B3)| < 0.001$) between glint affected and neighboring glint-free pixels for all open ocean (BRS, HFA) and two inland water (LCE) sample scenes, indicating that the sun glint contribution to TOA reflectance has been effectively removed. On four occasions, $|\Delta REF(B3)| > 0.001$ were observed for inland water sample scenes (LCE, LPY). This could indicate that sun glint is sometimes correlated with high water-leaving radiance, e.g., around river plumes characterized by highly turbid waters which prominently feature in both AOIs.

Under similar atmospheric conditions, GRCM should provide similar *TSGC* values for different images of a given AOI. The observed good agreement in TSGC for a number of cases (e.g., LCE-2 vs. LCE-3; LPY-1 vs. LPY-2) therefore hints to the robustness of the retrieval scheme. Due to reduced atmospheric Rayleigh scattering, TSGC values are significantly higher for the two high-mountain (LPY) sample scenes, especially in the VIS channels.

In two cases (HFA-2 and HFA-4), extreme glint coverage ($GAA > 99\%$) is associated with high values of the SWIR reflectance at TOA ($\rho^*_{AER}(B7) > 0.01$), leading in one case (HFA-2) to a spectral dependence of TSGC with an implausible maximum value in channel B3. Application of GRCM at very high glint coverage ($GAA > \sim 95\%$) is often problematic and should be excluded in automated processing.

GRCM-derived average *TSGC* values derived from the nine oceanic sample scenes (i.e., AOIs BRS and HFA) have been compared to physically equivalent average "slope" values derived by [23] for 227 OLI scenes taken over coastal waters offshore French Guiana. Both methods show good agreement for OLI channels B3, B4, and B6, while some larger differences are observed for channel B2 (where the atmospheric impact is largest) and to a lesser degree also for channel B5 (showing the lowest SNR). While these results indicate

the principal suitability of either correction scheme, a systematic analysis of strengths and weaknesses of the individual approaches beyond anecdotal evidence is deemed beyond the scope of the present work as this would require a careful setup involving significant community effort such as, for example, demonstrated through the Atmospheric Correction Intercomparison Exercise (ACIX-Aqua) [38].

**Table 4.** GRCM retrieval results for the OLI sample scenes listed in Table 3. Atmospheric surface pressure *(p_srf)* has been taken from ERA5 reanalysis [34], the solar zenith angle $\theta_S$ is taken from the OLI metadata file. The retrieval quality parameters $\rho^*_{AER}(B7)$, $\Delta AMRC$, and $\Delta REF$ are described in Section 3.4.2. Additionally shown is a comparison between average correction results for GRCM vs. the method of [23]. See text for further explanation.

| AOI-ID | p_srf [hPa] | $\theta_S$ [deg] | GAA [%] | $\rho^*_{AER}$ (B7) | $\Delta AMRC$ (B3) | $\Delta REF$ (B3) | c (B2) | c (B3) | c (B4) | c (B5) | c (B6) |
|---|---|---|---|---|---|---|---|---|---|---|---|
| BRS-1 | 1035 | 33.3 | 72 | 0.0044 | 0.00029 | 0.0003 | 0.71 | 0.92 | 0.98 | 1.00 | 1.04 |
| BRS-2 | 1017 | 45.6 | 100 | 0.0051 | 0.00109 | 0.0001 | 0.84 | 1.06 | 1.15 | 1.21 | 1.15 |
| BRS-3 | 1022 | 29.2 | 72 | 0.0031 | 0.00157 | 0.0005 | 0.72 | 0.96 | 1.06 | 1.14 | 1.16 |
| BRS-4 | 1014 | 36.9 | 78 | 0.0044 | 0.00039 | 0.0009 | 0.55 | 0.79 | 0.91 | 1.04 | 1.11 |
| HFA-1 | 1012 | 59.5 | 6 | 0.0030 | 0.00021 | 0.0002 | 0.55 | 0.76 | 0.83 | 0.91 | 0.90 |
| HFA-2 | 1015 | 30.7 | 100 | 0.0244 | 0.00091 | 0.0004 | 0.39 | 0.99 | 0.90 | 0.80 | 1.14 |
| HFA-3 | 1015 | 24.2 | 81 | 0.0054 | 0.00020 | 0.0004 | 0.61 | 0.87 | 0.89 | 0.89 | 0.95 |
| HFA-4 | 1016 | 22.0 | 99 | 0.0124 | 0.00037 | 0.0003 | 0.57 | 0.91 | 0.98 | 1.00 | 1.05 |
| HFA-5 | 1011 | 21.4 | 76 | 0.0080 | 0.00073 | 0.0006 | 0.85 | 1.02 | 1.11 | 1.17 | 1.12 |
| LCE-1 | 973 | 31.5 | 15 | 0.0031 | 0.00020 | 0.0008 | 0.46 | 0.65 | 0.78 | 0.96 | 1.09 |
| LCE-2 | 972 | 28.9 | 54 | 0.0008 | 0.00035 | 0.0008 | 0.75 | 0.97 | 1.04 | 1.09 | 1.10 |
| LCE-3 | 971 | 31.0 | 47 | 0.0006 | 0.00049 | 0.0029 | 0.74 | 0.99 | 1.08 | 1.15 | 1.16 |
| LCE-4 | 966 | 38.6 | 5 | 0.0007 | 0.00024 | 0.0018 | 0.68 | 0.93 | 1.05 | 1.11 | 1.12 |
| LPY-1 | 538 | 22.2 | 72 | 0.0054 | 0.00147 | 0.0027 | 1.09 | 1.19 | 1.25 | 1.25 | 1.13 |
| LPY-2 | 541 | 31.1 | 18 | 0.0003 | 0.00071 | 0.0055 | 1.09 | 1.17 | 1.22 | 1.22 | 1.10 |
| Average of 9 BRS and HFA open ocean sample scenes | | | | | | | 0.64 | 0.92 | 0.98 | 1.02 | 1.07 |
| Corresponding standard deviation | | | | | | | 0.15 | 0.10 | 0.11 | 0.14 | 0.09 |
| Average of 227 OLI scenes offshore French Guiana. Source: [23] | | | | | | | 0.83 | 0.90 | 0.99 | 1.08 | 1.09 |
| Corresponding standard deviation | | | | | | | 0.15 | 0.14 | 0.07 | 0.13 | 0.11 |

### 4.3. Additional Aspects

GRCM assumes horizontally homogeneous atmospheric conditions which may lead to artefacts due to local under- or overcorrection where the assumption does not hold. This issue may eventually be addressed by subdividing the area of interest into smaller subareas, but this in turn is limited by the need to have glint-free pixels in every subarea.

GRCM currently employs a relatively simple contrast measure to separate sun glint from background. Involving advanced pattern recognition methods (e.g., edge detection) might have a potential to further improve sun glint identification and quantification, especially under low-glint conditions.

In the current implementation, the SWIR aerosol reflectance at TOA and TSGC are determined sequentially. A two-dimensional minimization approach to determine both parameters simultaneously may prove more accurate, especially for scenes characterized by high glint cover where the SWIR TOA aerosol reflectance may be overestimated due to residual glint outside the GAA.

The implementation of GRCM is not overly complicated when using a programming language offering advanced array manipulation support such as IDL or Python/NumPy, for example, MRC can be comfortably calculated using grayscale erosion. The core of the method, i.e., the determination of the GAA and the application of the contrast minimization procedure, comprises just a few hundred lines of code. However, contrast minimization makes the operation of GRCM more time consuming than the Hochberg/Hedley-like regression schemes: in its current non-optimized implementation, the processing of e.g.,

sample area LCE-2 (ca. 800 × 1200 pixel) takes about 120 s on a basic Linux workstation (Intel® Core™ i5-10400 CPU, 40 GB RAM). While this is sufficient to execute case studies involving a limited number of OLI scenes, operational application of GRCM to large image quantities will require the implementation of optimized minimization schemes.

Finally, the approach to separate processes based on differing morphological characteristics is not limited to glint correction. A potential application could concern the correction of thin cirrus over contrasted surfaces—involving contrast maximization in this case.

### 4.4. Can Sun Glint Contribute to Atmospheric Correction?

As stated in Section 2.5, TSGC inherently contains information on the spectral dependence of the atmospheric transmittance relative to the chosen SWIR reference wavelength. This is reflected by the increasing differences in the TSGC values towards shorter wavelengths between LPY and the other AOIs, the former characterized by a significantly lower Rayleigh optical depth due to its high-altitude location at 5013 m above MSL and a correspondingly higher atmospheric transmittance (Table 4).

Expressing atmospheric transmittance as a function of optical depth and relative airmass, Equation (20) can be transformed to provide information on the difference in the aerosol optical depth (AOD) between OLI channel $B_n$ and reference channel $B_7$:

$$\Delta AOD := AOD\,(B_n) - AOD\,(B_7) = -\frac{\ln\left(\frac{c}{\varepsilon} \times \alpha\right)}{\left(\frac{1}{\mu_S} + \frac{1}{\mu_O}\right)}, \tag{32}$$

where the ratio $\alpha$ of the two-way Rayleigh transmittance in channel $B_7$ over channel $B_n$ can be calculated from the atmospheric pressure [32], and $\varepsilon$ is the spectrally normalized BRDF of the water surface as calculated and tabulated by [13].

Equation (32) has been tentatively applied to sample scene BRS-3. The results shown in Table 5 indicate a stronger spectral dependence for the GRCM-derived AOD as compared to the corresponding daily averages from the nearby AERONET station Brest_MF [39], the latter linearly interpolated to the central wavelengths of the OLI channels. For example, the AOD difference between channels B2 and B5 amounts to 0.12 for the AERONET observations, but to 0.39 if derived using Equation (32). There are several possible reasons for the observed discrepancies: the daily averaged AERONET AOD may not be representative of the conditions during the time of the OLI overpass, the conditions at the land based AERONET site may not be representative of the conditions above the nearby ocean, or GRCM may not have determined TSGC with sufficient accuracy.

**Table 5.** Estimation of spectral dependence of the aerosol optical depth according to Equation (32) for sample scene BRS-3. See text for further explanation.

|  | **B2** | **B3** | *B*4 | *B*5 | *B*6 |
|---|---|---|---|---|---|
| $c$ | 0.72 | 0.96 | 1.06 | 1.14 | 1.16 |
| $\varepsilon$ | 1.27 | 1.25 | 1.23 | 1.21 | 1.13 |
| $\alpha$ | 0.70 | 0.83 | 0.90 | 0.97 | 1.00 |
| $\Delta AOD$ | 0.43 | 0.21 | 0.12 | 0.04 | −0.01 |
| AERONET $AOD$ | ~0.22 | ~0.18 | ~0.15 | ~0.10 | ~0.07 |

An analysis beyond the scope of the present work is required to assess the potential of GRCM to provide useful information for atmospheric correction purposes. At this point, it can just be concluded that sun glint at TOA is not only a source of noise that needs to be corrected in order to extract the sub-surface signal with reasonable accuracy, but also constitutes a potentially valuable source of information on spectral atmospheric properties.

### 5. Conclusions

A novel sun glint correction scheme for high spatial resolution (≤50 m) imagery has been established, exploiting the sun glints' morphological characteristics occurring at

such spatial scale. The scheme implements a contrast minimization approach to isolate and subsequently remove the sun glint contribution from the TOA reflectance at VNIR wavelengths using the sun glint pattern extracted from concomitant SWIR observations.

The scheme, termed GRCM (Glint Removal through Contrast Minimization), has been applied with good success to a suite of 15 OLI scenes encompassing a wide range of environmental conditions: glint corrected images reveal a lot of in-water and underwater features not or only faintly visible in the glinted images. Glint corrected images are showing only minor correction-induced artefacts, pointing to a good numerical stability of the underlying minimization approach.

The quantitative evaluation of GRCM has proven more challenging: while relatively good agreement of the GRCM-derived TSGC values with, e.g., those published by [23] for an area offshore French Guiana is observed, such comparison is inherently anecdotical and would need to be extended in the context of a systematic intercomparison exercise such as e.g., ACIX-Aqua [38]. In such context, the potential of the retrieved TSGC values for the characterization of the spectral AOD relative to a SWIR wavelength could be analyzed as well.

**Funding:** This research received funding from the European Space Agency under contract 4000115822/15/I-SBo (SEOM S2-4Sci), the European Union's Horizon 2020 Research and Innovation programme under grant agreement No. 773421 (Nunataryuk), and the Israel Oceanographic and Limnological Research Ltd. under Purchase Orders PO22001889 and PO22001891.

**Data Availability Statement:** Landsat 8 Collection 2 Level 1 Tier 1 data courtesy of the U.S. Geological Survey. ERA5 hourly data on single levels from 1979 to present [38] was downloaded from the Copernicus Climate Change Service (C3S) Climate Data Store. The GRCM code can be made available upon request.

**Acknowledgments:** The author would like to thank three anonymous reviewers whose well-founded and thoughtful comments and suggestions helped to significantly improve an earlier version of the manuscript.

**Conflicts of Interest:** The author declares no conflict of interest.

## Abbreviations

| Acronym/Subscript | Explanation |
|---|---|
| ACIX | Atmospheric Correction Intercomparison Exercise |
| AER | Aerosol |
| AERONET | Aerosol Robotic Network |
| AMRC | Average maximum reflectance contrast |
| AOD | Aerosol optical depth |
| AR | Aerosol-Rayleigh (coupling term) |
| AOI | Area of interest |
| BRDF | Bidirectional reflectance distribution function |
| BGT | Bright (pixel) |
| BRS | Brest AOI (France) |
| BUF | Buffer |
| CLD | Cloud |
| COR | (Glint) Corrected |
| ERA5 | ECMWF Reanalysis 5th Generation |
| GAA | Glint affected area |
| GAP | Glint affected pixel |
| GRCM | Contrast Removal through Contrast Minimization |
| HFA | Haifa Bay AOI (Israel) |
| L1TP | Level 1 Terrain Precision |
| LCE | Lake Constance East AOI (Germany, Austria, Switzerland) |
| LPY | Lake Puma Yumco AOI (China) |

| MERIS | Medium Resolution Imaging Spectrometer |
|---|---|
| MODIS | Moderate Resolution Imaging Spectroradiometer |
| MRC | Maximum reflectance contrast |
| MSI | Multi-Spectral Imager |
| MSK | Mask (image) |
| MSL | Mean sea level |
| NDWI | Normalized difference water index |
| NIR | Near infrared, wavelength range ca. 0.7–1.5 μm |
| OLCI | Ocean, Land and Cloud Imager |
| OLI | Operational Land Imager |
| PGP | Potentially glinted pixel |
| POLYMER | Polynomial based algorithm applied to MERIS |
| PPRC | Pixel-to-pixel reflectance contrast |
| RAY | Rayleigh |
| SHD | (Cloud) Shadow |
| SMAC | Simplified Method for Atmospheric Correction |
| SUG | Sun glint |
| SWIR | Shortwave infrared, wavelength range ca. 1.5–2.5 μm |
| THR | Threshold |
| TOA | Top-of-atmosphere |
| TSGC | TOA Spectral Glint Conversion |
| VIS | Visible, wavelength range ca. 0.4–0.7 μm |
| VNIR | Visible and near infrared, wavelength range ca. 0.4–1.5 μm |
| WAT | Water |
| WCP | White caps |

## Appendix A

Appendix A contains examples of the GRCM performance for three sample scenes. To allow for direct comparison, visual presentation is identical for all scenes.

*Appendix A.1 Sample Scene HFA-5: Haifa Bay, 11 June 2022*

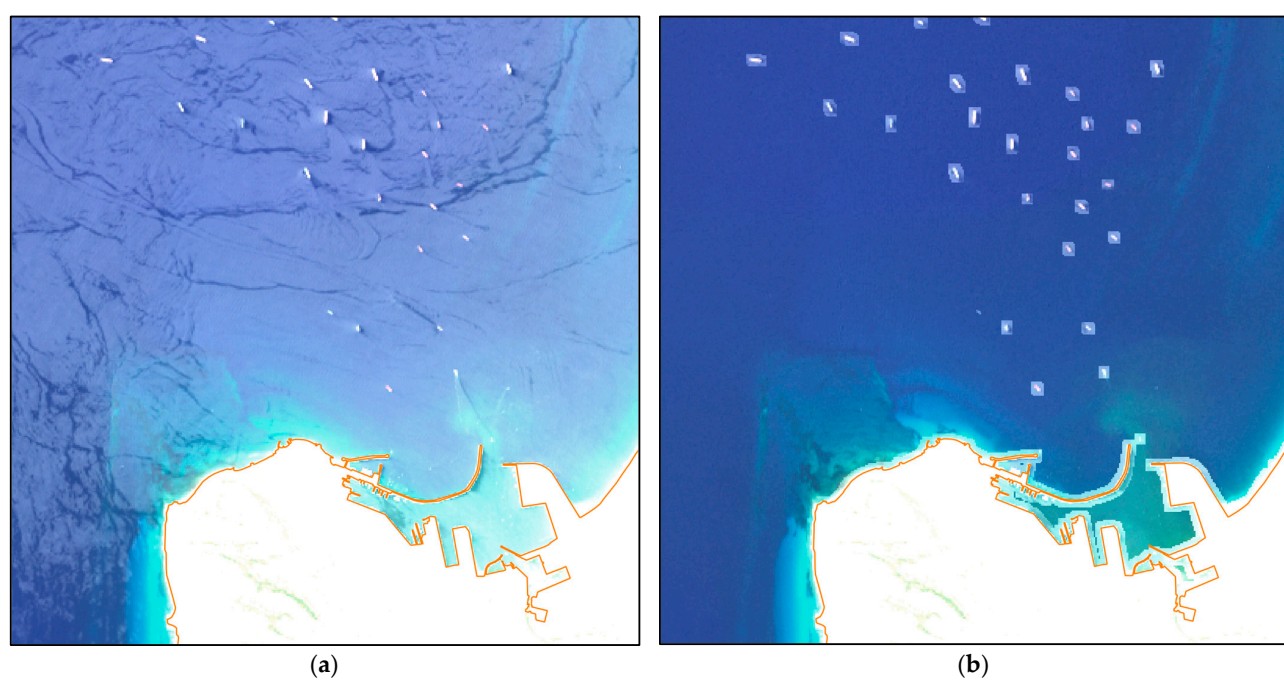

(a)                                                                  (b)

**Figure A1.** *Cont.*

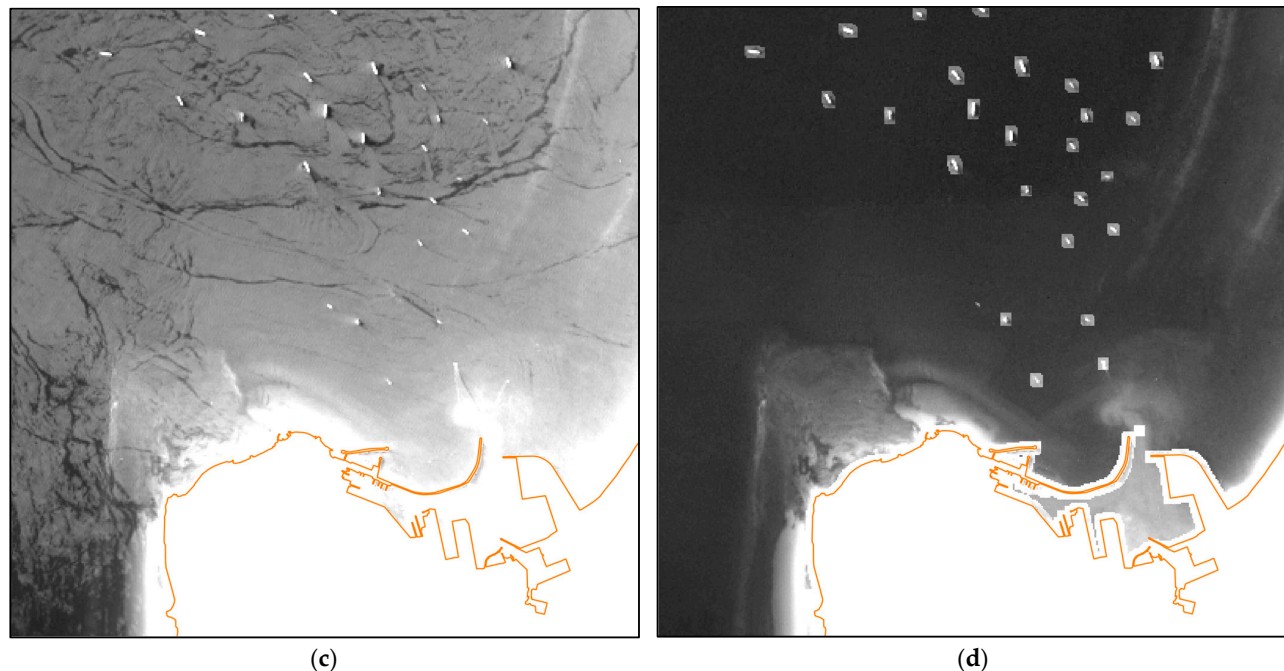

**Figure A1.** $12 \times 12$ km$^2$ subarea of sample scene HFA-5, Haifa Bay. (**a**) RGB image with glint; (**b**) RGB image glint corrected; (**c**) OLI B3 reflectance with glint; (**d**) OLI B3 reflectance, glint corrected. Although HFA-5 is strongly glint contaminated, GRCM provides visually good results: both the turbid plume off the entry to the harbor basin and the offshore bathymetric features in the lower left and upper right quadrants have become clearly visible.

*Appendix A.2 Sample Scene LCE-2: Lake Constance East, 1 June 2020*

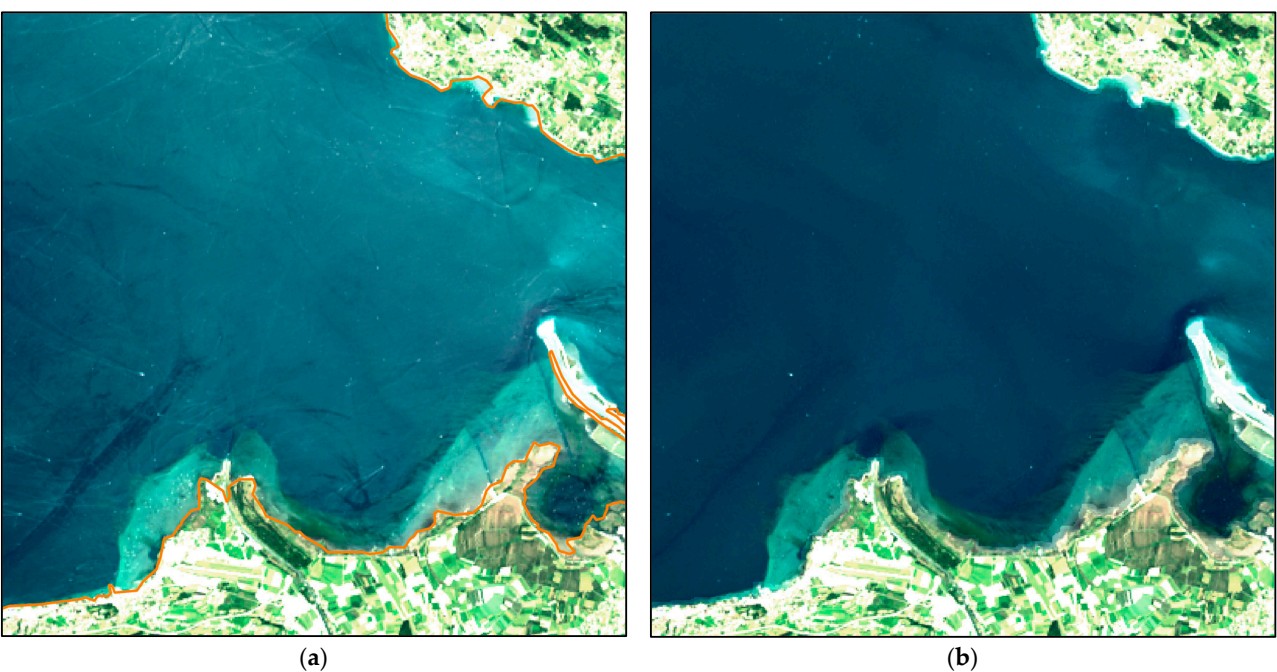

**Figure A2.** *Cont.*

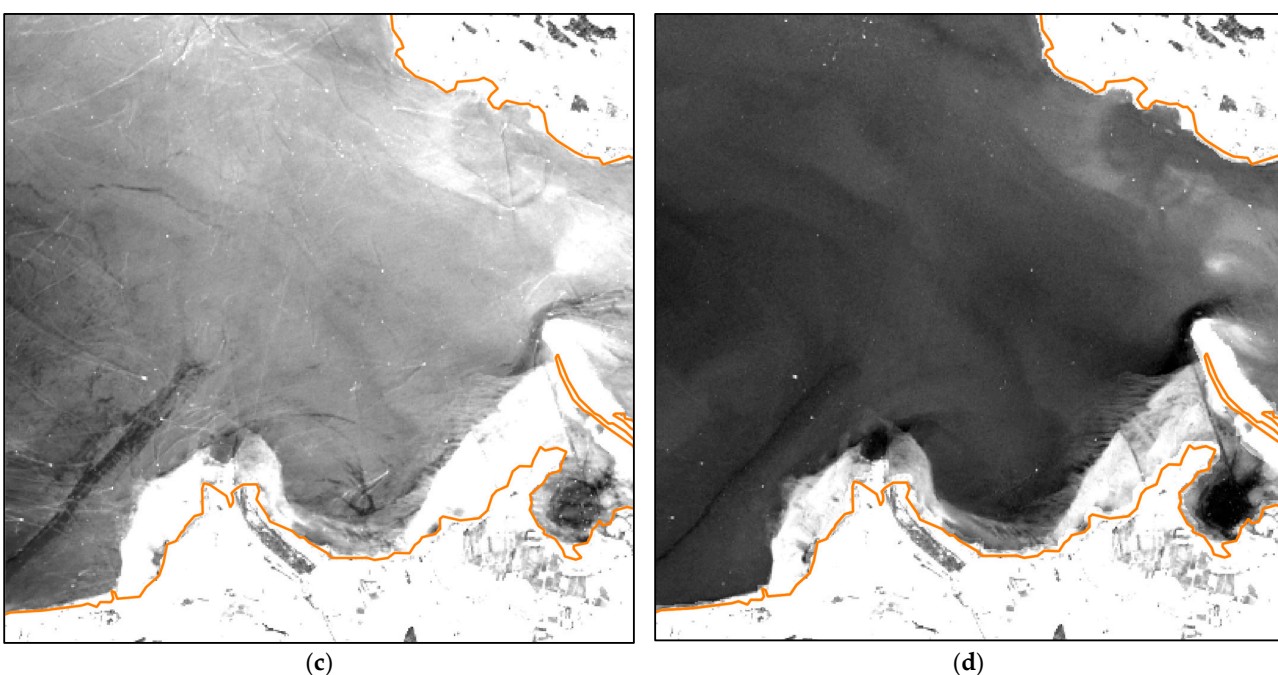

(**c**)                                                          (**d**)

**Figure A2.** $12 \times 12$ km$^2$ subarea of sample scene LCE-2, Lake Constance East. (**a**) RGB image with glint; (**b**) RGB image glint corrected; (**c**) OLI B3 reflectance with glint; (**d**) OLI B3 reflectance, glint corrected. While LCE-2 is rather mildly glint contaminated, there are many small leisure boats not identified as bright objects which may have had a negative impact on GRCM. For example, the long dark diagonal structure in the lower left quadrant of (**d**) may constitute a correction artefact. Still, glint correction is deemed successful from visual inspection: dynamic processes in the water body (e.g., turbid plumes) can be clearly identified, the bathymetric features along the southern shore can be better delineated, and most of the boat wakes have been removed.

*Appendix A.3 Sample Scene LPY-1: Lake Puma Yumco, 6 July 2018*

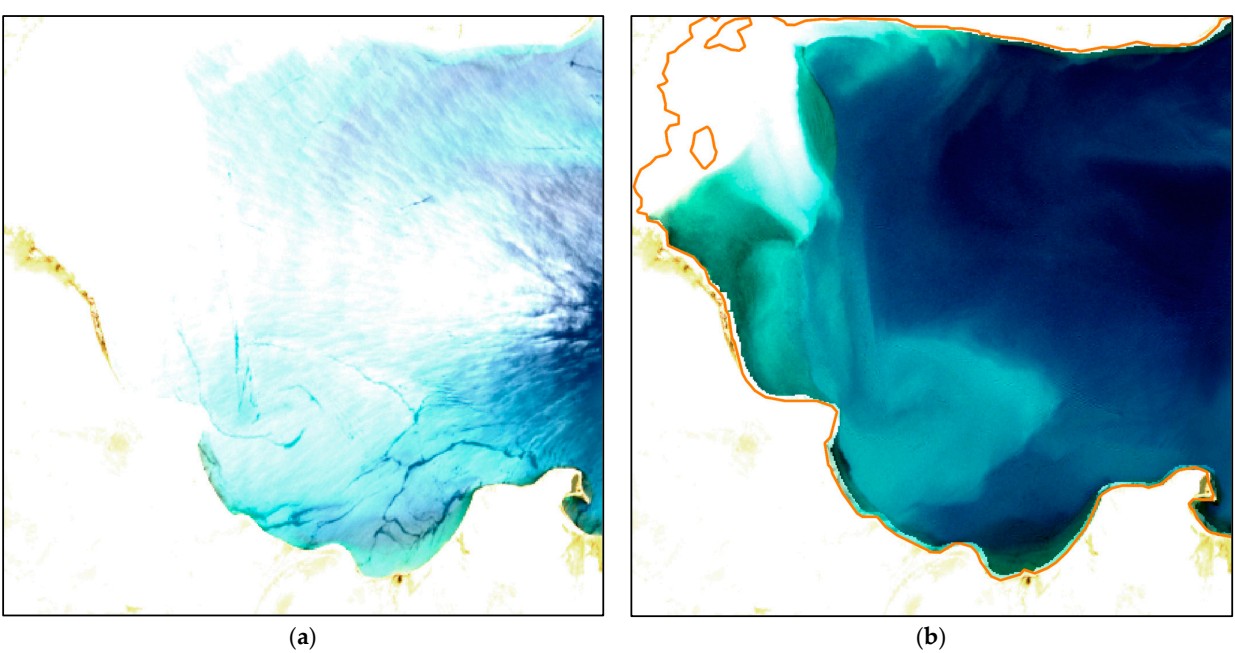

(**a**)                                                          (**b**)

**Figure A3.** *Cont.*

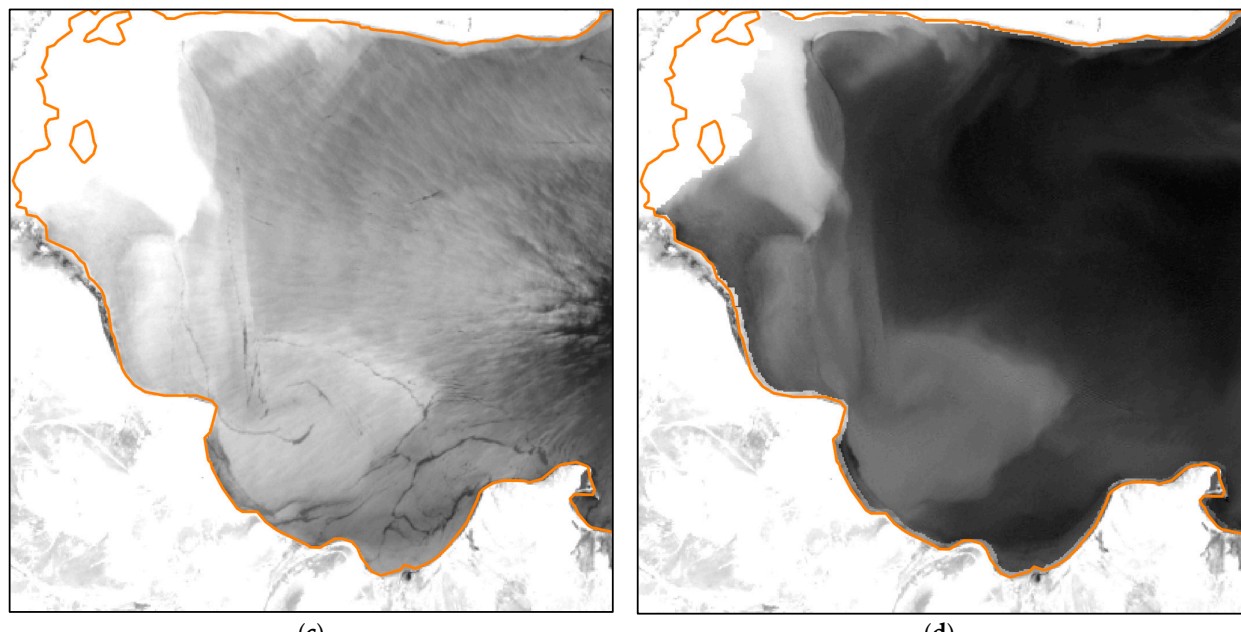

(**c**) (**d**)

**Figure A3.** $12 \times 12$ km$^2$ subarea of sample scene LPY-1, Lake Puma Yumco. (**a**) RGB image with glint; (**b**) RGB image glint corrected; (**c**) OLI B3 reflectance with glint; (**d**) OLI B3 reflectance, glint corrected. LPY-1 is among the heaviest glint contaminated sample scenes but comprises still sufficiently large glint-free areas to allow for an accurate determination of the SWIR TOA reflectance. The upper left quadrant is characterized by the influx of very turbid water; partly masked out through bright pixel masking (Section 3.2.2). Elsewhere, glint correction provided good results even in turbid waters: both in-water dynamic processes and some bathymetric features are clearly discernible. A few minor artefacts have been produced at the eastern tip of the triangular turbidity plume in the lower right quadrant.

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
