# Peer review of "A Contrast Minimization Approach to Remove Sun Glint in Landsat 8 Imagery"

_remotesensing, doi:10.3390/rs14184643_

Round 1

Reviewer 1 Report

This paper describes a new method for correcting sun glint in high resolution satellite imagery, demonstrated for the OLI instrument on Landsat 8. The method uses information from infrared bands to correct glint in the visible bands, as has been done before, but uses a novel approach to identify glint-affected pixels by exploiting the observation that glint leads to much greater pixel-to-pixel variation in reflectance than is found in areas without glint. As it stands the method is quite labour-intensive and more work is needed to create an optimised method ready for operational use. However, the method is novel and interesting - well worth publication in its current form. The paper is thorough and well presented, and I recommend it for publication after a few minor adjustments.

My main suggestion is to include more images. As the author states (lines 498-499), visual inspection can be used for a quick qualitative assessment of the success of glint correction. But this is shown for only one of the images used, and only for one band (Figure 7d). A set of RGB images for a few contrasting scenes, before and after correction, would give the reader a quick way to assess what the new method can achieve in different kinds of conditions. There is performance information in Table 4, but images would supplement and enrich this.

The paper describes the method in full detail, including the limitations of the method and further work needed. I have only a few comments on the text.

Section 2.2 gives some background on sun glint, but it does not seem to relate specifically to the method presented here and I was not sure why it is included in the Materials and Methods section – it could be moved to the introduction and perhaps shortened. Similarly I wasn’t sure why section 2.3 is included in the methods, except that it is related to the choice of thresholds (in section 2.3.3). If section 2.3 is included I suggest that it needs to be motivated more clearly – at the moment it feels like a diversion, before section 2.4 gets back to the core work of the paper. The rest of section 2 describes the method in full, then its implementation is demonstrated in section 3. The text here is quite dense, which is difficult to avoid given the level of detail needed, and it might be helpful to include references to the relevant parts of section 2 in the flow chart (Figure 5). A summary of the main points at the end of section 2 would also help to orientate the reader before moving on to section 3.

Section 3.8 lists a few ways in which the quality of the glint correction can be assessed. One approach that is not mentioned would be to apply the method to a scene where in situ measurements of irradiance are available, as is done in reference [4], for example. Was this an option that the author considered?  

Line 573 refers to agreement in the retrieved TSGC as an indication of the robustness of the retrieval scheme.  I am not clear why this is - why would we expect the TSGC to be consistent between images with different atmospheric and water surface conditions?

Minor corrections:

Line 430 “in” is missing – “is depicted in figure 6”.

On line 497 [H] is written at the end of the line, whereas in the other section headings the letter is at the start.

I am not familiar with the convention “resp.” used on lines 119, 141, 223 and 363.

Reviewer 2 Report

Sun glint is commonly present in the satellite imagery particularly in the high spatial resolution data. Efficient removal sun glint is a prerequisite when using high resolution data in quantitative analysis such as in chlorophyll concentration retrieval. This manuscript proposed a new method to identify and remove the sun glint in the OLI imagery by using a contrast minimization algorithm. This method could be extended to other high spatial resolution data such as MSI or Worldview. I would recommend this manuscript to be published after addressing the following issues. 

1.     I don’t think section 2.3 is necessary. The negative effects of sun glint to the retrieval of water constituent are well known. The amount of effects depends on not only the sun glint severity but also the retrieval method. For example, baseline subtraction algorithm such as color index (Hu et al. 2012)will be less affected by the sun glint. 

2.     Many imaged-based sun glint correction methods had been published in the literature. The proposed is rather complicated to be implemented. The author should clearly point out the advantage of proposed new method and compare the performance with other commonly used simple sun glint correction method such as Hedley et al. (2005)

3.     Figure 6 quantitively showed the performance of the proposed glint removal method. However, more quantitative analysis is needed such as the reflectance comparison with sun glint free area or the in situ measurements. 

4.     Many thresholds were determined in the manuscript. Are these thresholds applicable to other area or other sensors? The author needs to clarify it. 

5.     In section 2.4.2, how pixel-to-pixel reflectance contrast (PPRC) is calculated? Is the median/mean/maximum reflectance difference of center pixel with its neighbors?

6.     The abbreviations should be present when it appears in the first time in the main text. For example, GAA need to be abbreviated in Line 245 not in Line 285 and it is abbreviated again in Line 435. Please check other abbreviations. 

7.     In Figure 3a, 3b, 7a and 7b, there is [1] in the label of y-axis. Please explain it. 

8.     In the title of section 3.8, [H] should be at the beginning not the end of the title. Also, why only in section 3.1, square brackets embraced capital letter appears in the title of each sub-section but not in other sub-sections. 

9.     In Line 574, PYC-1 and PYC-2 should be a typo of LPY-1 and LPY-2 shown in Table 4.

10.   It's better to have a conclusion section to emphasis the contribution of this manuscript. 

Hedley, J.D., Harborne, A.R., & Mumby, P.J. (2005). Technical note: Simple and robust removal of sun glint for mapping shallow-water benthos. International Journal of Remote Sensing, 26, 2107 - 2112

Hu, C., Lee, Z., & Franz, B. (2012). Chlorophyll a algorithms for oligotrophic oceans: A novel approach based on three-band reflectance difference. J. Geophys. Res., 117, C01011

Reviewer 3 Report

1. Introduction (line 81) ; please discuss the main weakness of the previous works and points of distinction of this study

2. Materials and method (line228-233, 254-255) : Could the authors mark the part in the figure they are discussing? I couldn't properly identify the locations in the figure they mentioned.

3. Materials and method (line256-259) : How about include a B7 (SWIR) band image? It might be helpful for readers to easily understand the author's discussion. By the way, I saw bigger differences in the B5 (red) band than in the B7 from the Fig. 3b.

4. Materials and method (line262-266) : I'm not sure how the author's could say that.

5. Materials and method (eqs. 6, 7) : More detailed explanations are needed.

6. Practical impletation (Figure 5) : I think the flow chart is a good way of explaining the process of the study. If the author can add the key word of each step discussed in the previous section (2.Materials and method) corresponging to each step of the flow chart, it would be more helpful for readers to intuitively understand their works. 

7. Practical impletation (line487-490) : More detailed discussion including some examples for specific locations in the figure is needed.

Round 2

Reviewer 2 Report

The author has address all of my concerns. 

Author Response

Reviewer 2 raised no further specific comments.